# Neovascular Age-Related Macular Degeneration: Therapeutic Management and New-Upcoming Approaches

**DOI:** 10.3390/ijms21218242

**Published:** 2020-11-03

**Authors:** Federico Ricci, Francesco Bandello, Pierluigi Navarra, Giovanni Staurenghi, Michael Stumpp, Marco Zarbin

**Affiliations:** 1Department of Experimental Medicine, University Tor Vergata, Viale Oxford, 00133 Rome, Italy; 2Scientific Institute San Raffaele, University Vita Salute, 20132 Milan, Italy; bandello.francesco@hsr.it; 3Fondazione Policlinico Universitario A. Gemelli IRCCS, 00168 Rome, Italy; Pierluigi.Navarra@unicatt.it; 4Department of Pharmacology, Medical School, Catholic University, 00198 Rome, Italy; 5University Eye Clinic Luigi Sacco Hospital, 20157 Milan, Italy; giovanni.staurenghi@unimi.it; 6Molecular Partners AG—Wagistrasse, 14 8952 Zurich-Schlieren, Switzerland; michael.stumpp@molecularpartners.com; 7Institute of Ophthalmology and Visual Science, Rutgers-New Jersey Medical School, Newark, NJ 07103, USA; zarbin@earthlink.net

**Keywords:** age-related macular degeneration, neovascular AMD, neovascularization, vascular endothelial growth factor, anti-VEGF, Ang-2, DARPins

## Abstract

Age-related macular degeneration (AMD) constitutes a prevalent, chronic, and progressive retinal degenerative disease of the macula that affects elderly people and cause central vision impairment. Despite therapeutic advances in the management of neovascular AMD, none of the currently used treatments cures the disease or reverses its course. Medical treatment of neovascular AMD experienced a significant advance due to the introduction of vascular endothelial growth factor inhibitors (anti-VEGF), which dramatically changed the prognosis of the disease. However, although anti-VEGF therapy has become the standard treatment for neovascular AMD, many patients do not respond adequately to this therapy or experience a slow loss of efficacy of anti-VEGF agents after repeated administration. Additionally, current treatment with intravitreal anti-VEGF agents is associated with a significant treatment burden for patients, caregivers, and physicians. New approaches have been proposed for treating neovascular AMD. Among them, designed ankyrin repeat proteins (DARPins) seem to be as effective as monthly ranibizumab, but with greater durability, which may enhance patient compliance with needed injections.

## 1. Introduction

Age-related macular degeneration (AMD) constitutes a prevalent, chronic, and progressive retinal degenerative disease of the macula that affects elderly people and causes central vision impairment as a result of damage to retina, retinal pigment epithelium (RPE), and choriocapillaris [1,2,3].

AMD constitutes one of the leading causes of irreversible visual impairment in developed countries [4,5,6,7,8,9,10,11,12,13]. Its overall prevalence is approximately 8.7%, although variation among different populations is substantial [4,5,6,7,8,9,10,11,12,13]. The results of a metanalysis that included 129,664 subjects, with 12,727 cases from 39 studies, showed that the prevalence of AMD ranged from 7.3% in Asian populations to 12.3% in European ancestry populations [4]. Such prevalence rate variations among the different populations may be explained by genetic differences, lifestyle, and/or dietary factors [14,15].

Due to the increase in life expectancy, it might be assumed that AMD will become more prevalent [16]. The metanalysis estimated the number of patients affected by AMD worldwide to be 288 million by 2040 [4].

Additionally, according to the results of another metanalysis, which included 42,080 subjects from 10 European countries, the overall prevalence of AMD among subjects ≥70 years was 13.2% and 3.0% for early and late AMD, respectively [17]. Moreover, in Europe, late AMD is expected to affect 77 million people by 2050 [18].

Regarding early AMD stages, the prevalence rates in Europe (15.4% to 29.5%) and in North America (14.1% to 20.0%) are comparable, although greater than in Asia (3.1% to 13.9%) [4,19]. Nevertheless, despite these differences in the overall and early AMD prevalence rates, the prevalence of late AMD is similar in European (2.2% to 2.5%), North American (1.1% to 2.1%), and Asian (0.1% to 1.9%) studies [4,19].

The RPE is crucial for the maintenance of photoreceptor cells and is involved in recycling of visual pigments and daily phagocytosis of photoreceptor outer segments *inter alia*. With aging, the RPE suffers diverse changes associated with the emergence of deposits of abnormal extracellular material, drusen, that can be detected clinically in the area centralis as well as in the fundus mid periphery. Histologically, typical drusen lie at the interface between the RPE and the inner collagenous zone of Bruch’s membrane, and reticular pseudodrusen are located between the RPE and photoreceptors [1,2,20,21].

Generally speaking, AMD can be classified as early, intermediate, or late [22]. Late stage AMD is associated with the development of geographic atrophy (atrophy of the photoreceptors and subjacent RPE and choriocapillaris) and choroidal neovascularization, which are not mutually exclusive outcomes [23]. Previously, wet AMD was responsible for most of the severe vision loss and usually occurs over weeks to months. Wet AMD has been subclassified as Type 1 “occult”/polypoidal choroidal vasculopathy, Type 2 “classic”, and Type 3 “retinal angiomatous proliferation” [24]. The angiogenesis and increased vascular permeability seen in wet AMD is driven partly by upregulation of vascular endothelial growth factor (VEGF) [2,20,21,22,23].

Angiogenesis is a multistep, tightly regulated process that is controlled by a dynamic balance of positive and negative factors. It has been proposed that the main type responsible for angiogenesis is VEGF-A, interacting with VEGF receptor (VEGFR) 1 and 2 [25].

The availability of anti-vascular endothelial growth factor (anti-VEGF) therapy has changed the prevalence of this condition substantially [26]. Despite having drugs that achieve positive results in clinical trials, real-world results differ significantly [27]. Moreover, clinical trials have also shown poor long-term results of anti-VEGF [28].

Newer treatment strategies are, therefore, needed to close the gap between clinical trial results and real-world settings. Currently, there are several therapeutic options that are being investigated—designed ankyrin repeat proteins (DARPINs), brolucizumab, use of port-delivery systems (PDS), and bispecific antibodies.

This study aims to provide an overview of our current understanding of the neovascular AMD (NVAMD), focusing on its current management and its future trends.

## 2. Pathophysiology

Despite intensive basic and clinical research, AMD pathogenesis has not been fully elucidated, likely due to its multifactorial character [20,29,30,31].

With the exception of smoking, no consistent evidence indicates that modifiable factors such as lipid levels, blood pressure, light exposure, or alcohol intake put people at substantially greater risk of AMD [1,2,20,29,30,32,33].

The RPE is crucial for the maintenance of photoreceptor cells. It promotes a vascular environment along its basal surface and an avascular environment along its apical surface [34]. With aging, several changes happen in the RPE cells as a result of their capacity for removing residual substances such as lipofuscin [35]. There is strong evidence that complement abnormalities play a central role in the pathogenesis of AMD, with mutations in the *complement factor H* gene increasing the risk of AMD 2.7–7.4-fold [36,37,38,39,40]. Various components of complement factor *C3* are present in the subRPE space, in drusen, and in the choroid of AMD eyes, and the terminal complement component, *C5*, has been localized in drusen and in the subRPE regions of AMD eyes post-mortem [41]. Soft drusen contain *C3a* and *C5a* and induce the upregulation of VEGF in RPE, which is consistent with the increased risk of choroidal neovascularization (CNV) associated with soft drusen [42]. Also, membrane attack complex (*C5b-9*, *MAC*) is localized in drusen and in compromised RPE cells of AMD eyes [41]. Additionally, drusen contain amyloid-β, which can activate the alternative complement pathway [43,44]. The complement system also is continuously activated in the eye [45].

Oxidative damage, which accompanies aging, can compromise regulation of the complement system by RPE cells. Oxidative stress reduces the surface expression of the complement inhibitors, decay accelerating factor (*CD55*) and *CD59*, and impairs complement regulation at the cell surface by factor *H* [46]. Oxidative stress interferes with the ability of interferon-γ to increase complement factor *H* expression in RPE cells, and products of photo-oxidation of *N*-retinylidene-*N*-retinylethanolamine (A2E) in RPE can trigger the complement system in vitro [47]. A preclinical in vivo model also links oxidative damage and complement activation to AMD [48]. The increased AMD risk associated with smoking is consistent with these findings. One theory of pathogenesis, consistent with histological and clinical observations, is that the natural history of AMD progresses from early stage disease (rod photoreceptor loss, medium-size drusen, Bruch’s membrane thickening) through intermediate stage disease (large confluent drusen, pigmentary changes) to advanced disease with GA in association with progressive choriocapillaris and RPE damage.

As a consequence of progressive RPE dysfunction, the permeability of the Bruch membrane is impaired, and therefore substances that would usually be removed by the choriocapillaris accumulate between RPE and Bruch membrane [49,50], resulting in the emergence of drusen [51,52]. Drusen are considered the hallmark of AMD.

The appearance of drusen is associated with thickening of the collagenous layers of Bruch’s membrane, which, in turn, is associated with lipid and protein accumulation and increased advanced glycation end product formation [3,53,54,55,56,57,58]. These changes, which are the result of metabolic dysfunction, may contribute to AMD progression by compromising the passage of nutrients between the choroid and outer retina as well as by creating a hypoxic environment, the consequence of chronic inflammatory damage to the choriocapillaris.

Retinal and choroidal hypoxia may induce an upregulation of VEGF production by the RPE, and thus promote the growth of abnormal vessels from the choroid (primarily) and retina (less often) [59,60,61].

The most studied factor related to ocular neovascularization is VEGF. The VEGF gene encodes a family of glycoproteins, whose primary function is the formation of blood vessels de novo (vasculogenesis, as occurs in embryonic development) and angiogenesis (formation of new blood vessels from preexisting vessels) by activating cellular signal pathways [62].

The VEGF family includes VEGF-A, VEGF-B, VEGF-C, VEGF-D, VEGF-E, and VEGF-F and placental growth factor (PlGF) [62,63].

We have evidence supporting the role of VEGF-A in vascular proliferation and migration of endothelial cells for both physiological and pathological angiogenesis [62]. Hypoxia induces the expression of VEGF-A and other pro-angiogenic factors to promote the formation of new vessels [64]. However, high expression of VEGF by RPE cells is not sufficient to cause CNV in some preclinical models [65]. Thus, CNV pathophysiology in AMD may involve more than hypoxia-induced increased in VEGF-A production.

Hypoxia-inducible factor-1 (HIF-1) is a heterodimer made up of HIF-1α and HIF-1β [66]. Hypoxia causes HIF-1α to accumulate and bind to HIF-1β to form HIF-1 [66]. Among others, genes that are transcriptionally regulated by HIF-1 include VEGF and its receptor VEGFR1; platelet-derived growth factor-B (PDGF-B) and its receptor PDGFRβ; stromal-derived factor-1 (SDF-1) and its receptor CXCR4; angiopoietin-2 (Ang2); and vascular endothelial-protein tyrosine phosphatase (VE-PTP) [67]. Angiogenesis is regulated by HIFs, with VEGF and the VEGFR major family as the principal intermediary [68].

In addition to secreted factors, which are primarily regulated by HIF-1, signals derived from the extracellular matrix and surrounding cells also are involved in CNV formation [34]. Oxidative stress has long been considered a major influence on the RPE in AMD pathophysiology. Vascular dysfunction may result in oxidative stress and overproduction of reactive oxygen species (ROS) [69]. In the fovea, the predominant photoreceptors are cones, which have higher energy consumption than rods, and therefore, higher demand for oxygen [70]. Moreover, cones are more susceptible to free radical-induced damage [71].

In NVAMD, therefore, CNV may be considered as a secondary reaction promoted by either stressor damage to the RPE or an immune response, which may result in hypoxia and up-regulation of angiogenic factors in the choroid, leading to the formation of pathological vessels [72,73,74]. In fact, neutrophils, macrophages, mast cells, activated microglia, all are capable of producing and releasing an array of pro-angiogenic factors, including VEGF [41,75,76]. Figure 1 shows different pathways involved in the pathophysiology of NVAMD.

As noted above, genetics is involved in the pathogenesis of NVAMD [77,78,79]. Various genetic variants, including complement factor *(CF)H* and *CFH-related genes* [80,81]; complement protein (C)3 and *C9* [82,83]; age-related maculopathy susceptibility (ARMS)2 gene [84]; and the VEGF and VEGF receptor (VEGFR) axis [85,86] are involved in AMD pathogenesis. Additionally, certain relationships between inhibitor metalloproteinase (TIMP) 3; fibrillin; collagen 4A3; and metalloproteinase (MMP) 19 and −9 [87] and NVAMD have been suggested.

## 3. Treatment Strategies of NVAMD

From a clinical perspective, NVAMD is characterized by CNV with intraretinal or subretinal leakage, hemorrhage, and RPE detachment [1,2]. Despite therapeutic advances in the management of NVAMD, none of the currently used treatments cures the disease or reverses its course.

### 3.1. Vascular Endothelial Growth Factor Inhibitors

Medical treatment of NVAMD experienced a significant advance due to the introduction of anti-VEGF agents. They dramatically altered the prognosis of the disease, which changed from “almost-certain” blindness [88] to a significant chance (~30%) of visual acuity (VA) improvement at least during the first two years of treatment [89,90,91].

Many different studies have evaluated the efficacy and safety of anti-VEGF in NVAMD patients. The results of these studies point on the same general direction, indicating a significant reduction in central macular thickness (CMT) and a visual acuity improvement. An overview of the different studies evaluating the efficacy of anti-VEGF in AMD patients has been summarized in Table 1 and Table 2.

#### 3.1.1. Pegaptanib

The first intravitreal anti-VEGF treatment developed specifically for NVAMD was pegaptanib sodium. Pegaptanib is a pegylated oligoribonucleotide (aptamer) that binds with high specificity and affinity to VEGF_165_, sequestering and therefore preventing VEGF_165_ from activating its receptor [123].

Two concurrent, multicenter, prospective, randomized, double-blind, and controlled clinical trials were conducted on AMD patients, at 117 sites in the United States, Canada, Europe, Israel, Australia, and South America [92]. These studies compared three pegaptanib intravitreal injection dosages (0.3 mg, 1.0 mg, and 3.0 mg) given every six weeks over a 48 week period versus (vs) sham injections. Among the 1208 patients randomly assigned to treatment, 1186 patients received at least one study treatment and were included in efficacy analyses [92]. As compared to sham injection (55%), a greater proportion of eyes with visual acuity (VA) loss < 15 letters was demonstrated for the 0.3 mg (70%, *p* < 0.001); 1.0 mg (71%, *p* < 0.001); and 3.0 mg (65%, *p* = 0.03) pegaptanib injections. Additionally, as compared to the sham injection group, a significantly higher proportion of patients maintained or improved their VA in the 0.3 mg (*p* = 0.003), 1.0 mg (*p* < 0.001), and 3.0 mg (*p* = 0.02) pegaptanib study groups [92].

However, due to its poorer efficacy compared with other currently available anti-VEGF drugs, pegaptanib is no longer recommended for the treatment of exudative AMD.

Regarding safety, incidence of ocular adverse events in the pegaptanib groups was greater than in the control group (see Table 3). Moreover, subjects in the pegaptanib groups were more likely to have a serious systemic adverse event than participants in the control one. Ocular adverse events are shown in Table 3.

#### 3.1.2. Bevacizumab

Bevacizumab, a humanized monoclonal antibody that inhibits VEGF-A, was originally developed as a medication for use in combination with existing metastatic colorectal cancer chemotherapy regimens [125,126].

Intravitreal injections of bevacizumab are used widely as an off-label treatment for NVAMD [127,128].

The ABC trial was a prospective, multicenter, double-masked, and randomized clinical trial (RCT) that analyzed the efficacy and safety of bevacizumab for treating NVAMD [96,97]. The results of this study showed that a significantly greater proportion of patients achieved best-corrected visual acuity (BCVA) improvement ≥15 letters from baseline in the bevacizumab group than in the control group (32% vs. 3%, respectively; *p* < 0.0001). Additionally, the mean BCVA improvement in the bevacizumab group was 7.0 letters, while in the standard of care group the mean BCVA decreased 9.4 letters (*p* < 0.001) [96]. At the week-54 examination, bevacizumab-treated patients were more likely to gain at least 6 letters or more of contrast sensitivity than the patients receiving standard care (PDT [photodynamic therapy], pegaptanib intravitreal injections, or sham treatment) (35.4% vs. 15.2%, respectively; *p* = 0.009) [97].

The CATT (Comparison of Age-Related Macular Degeneration Treatments Trials) study was a randomized, single-blind, and non-inferiority study (non-inferiority margin 5 ETDRS letters) that compare the efficacy and safety of bevacizumab and ranibizumab in patients with NVAMD [28,100,101]. In this study, bevacizumab was non-inferior to ranibizumab, both when the drugs were given monthly and when the drugs were given as needed (Figure 2) [100].

After two years, among those subjects who maintained the same treatment regimen, mean gain in VA was similar for both drugs (*p* = 0.21) [101]. However, as needed treatment posology resulted in less gain in VA (difference: −2.4 letters; *p* = 0.046) [104]. Interestingly, the 5-year results of the CATT study showed a decreased in BCVA (mean change −3 letters from baseline and −11 letters from 2 years), although 50% of eyes had a VA 20/40 or better [28].

The IVAN (alternative treatments to Inhibit VEGF in Age-related choroidal Neovascularization) study was a multicenter, randomized, and non-inferiority study that compared the efficacy and safety of bevacizumab (1.25 mg) and ranibizumab (0.5 mg) in patients with NVAMD. Subjects were randomly assigned to intravitreal injections of ranibizumab or bevacizumab in continuous (every month) or discontinuous (as needed) regimens, with monthly review [102,103]. In this study, the comparison of drugs was inconclusive. Using a 3.5 letter non-inferiority margin, bevacizumab was neither inferior nor equivalent to ranibizumab. Additionally, continuous treatment was equivalent to the discontinuous one [102]. The two-year results of the IVAN study confirmed these findings [103].

The aforementioned results are consistent with the results of the BRAMD study. This was a prospective, multicenter, randomized, controlled, and double-masked clinical trial that compared the effectiveness of bevacizumab and ranibizumab for treating NVAMD [105]. The mean BCVA improvement in the bevacizumab group (5.1 ± 14.1 letters) was similar to that observed in the ranibizumab group (6.4 ± 12.2); *p* = 0.37 [105]. Similarly, the GEFAL study, a multicenter, prospective, noninferiority, and double-masked RCT conducted in 38 French ophthalmology centers, found that after 1-year of follow-up bevacizumab was noninferior to ranibizumab in both the intent-to-treat and the per protocol analysis [106]. Finally, the MANTA study, was a multicenter, prospective, randomized, and parallel group study conducted in 10 Austrian centers [109]. At month 12, there was no significant difference in mean BCVA improvement between bevacizumab and ranibizumab (4.9 letters vs. 4.1 letters, respectively; *p* = 0.78) [109]. In summary, according to the currently available evidence, bevacizumab may be considered as a reasonable and affordable alternative to ranibizumab in patients with NVAMD [28,100,101,102,103,104,105,106,107,108,109,110,111,112,113,124,129].

Regarding safety, at one year, in the CATT 2011 [100], IVAN [102], and GEFAL 2013 [106] studies, less than 1% of participants had endophthalmitis, retinal detachment, retinal pigment epithelial tear, traumatic cataract, or uveitis. At two years, less than 1% of participants had endophthalmitis, retinal detachment, retinal pigment epithelial tear, traumatic cataract, or uveitis [101,103]. An overview of ocular adverse events is summarized in Table 4.

#### 3.1.3. Ranibizumab

Ranibizumab (Lucentis^®^, Novartis, Basel, Switzerland; Genentech, South San Francisco, CA, USA) is a fully humanized monoclonal antibody fragment that binds to multiple isoforms of VEGF-A [130]. It was originally approved for treating NVAMD [131].

The efficacy and safety of fixed regimens of ranibizumab were evaluated in various trials, including ANCHOR [90,93]; MARINA [89,91], PIER [94,95], and EXCITE [132]. ANCHOR was a prospective RCT that compared the efficacy and safety of ranibizumab (0.3 and 0.5 mg) vs. verteporfin therapy in patients with NVAMD [90,93]. Mean BCVA increased by 8.5 letters in the 0.3-mg group and 11.3 letters in the 0.5-mg group, as compared with a decrease of 9.5 letters in the verteporfin group (*p* < 0.001 for each comparison) (Figure 3).

Additionally, the results after two years of follow-up confirmed a clinically meaningful benefit of intravitreal ranibizumab vs. verteporfin therapy (34% to 41% patients treated with ranibizumab had gained ≥15 letters vs. 6.3% of them in the PDT group) [93].

MARINA (Minimally classic/occult trial of the Anti-VEGF antibody Ranibizumab in the treatment of Neovascular AMD) was a multicenter, 2-year, double-blind, and sham-controlled RCT designed to evaluate the efficacy and safety of intravitreal ranibizumab (0.3 mg and 0.5 mg) in patients with NVAMD [89,91]. At month 12, a significantly greater proportion of patients had lost fewer than 15 letters in the 0.3 mg ranibizumab group (94.5% of patients) or in the 0.5 mg ranibizumab group (94.6% of patients) than in the sham control group (62.2% of patients) with *p* < 0.001 for both comparisons [89]. Mean increases in visual acuity at month 12 were 6.5 letters in the 0.3 mg group and 7.2 letters in the 0.5 mg group, as compared with a decrease of 10.4 letters in the control group (*p* < 0.001 for both comparisons) [89] (Figure 4).

PIER was a multicenter, randomized, double-masked, and sham injection-controlled trial that evaluated the efficacy and safety of ranibizumab (0.3 mg and 0.5 mg) in patients with NVAMD treated quarterly after three monthly loading doses [94,95]. At month 12, mean changes from baseline VA were −16.3, −1.6, and −0.2 letters for the sham, 0.3 mg, and 0.5 mg groups, respectively (*p* ≤ 0.0001, for each ranibizumab dose vs. sham) [94]. At month 24, visual acuity VA had decreased an average of 21.4, 2.2, and 2.3 letters from baseline in the sham, 0.3 mg, and 0.5 mg groups (*p* < 0.0001 for each ranibizumab group vs. sham) [95]. It was suggested that in patients with NVAMD, retreatment need with intravitreal ranibizumab had a high intra-individual predictability, but that most patients required more frequent dosing than quarterly following the initial loading doses [133,134].

The efficacy of an individually tailored ‘observe-and-plan’ treatment regimen with ranibizumab was assessed in patients with NVAMD [135]. The results of this study indicated that mean VA improved by 8.7, 9.7, and 9.2 letters at months 3, 12, and 24, respectively. The mean treatment interval (after the loading doses) was 2.0 months during year-1 and 2.2 months during year-2 [135].

HARBOR was a 24-month, phase 3, randomized, multicenter, double-masked, and dose-response study. Subjects were randomly assigned in a 1:1:1:1 ratio to 1 of 4 ranibizumab treatment groups: 0.5 mg monthly, 0.5 mg pro re nata (PRN), 2.0 mg monthly, and 2.0 mg PRN. Mean change in BCVA (ETDRS letters) from baseline to month 12 was +10.1, +8.2, +9.2, and +8.6 for the ranibizumab 0.5 mg/monthly; 0.5 mg/PRN; 2.0 mg/monthly, and 2.0 mg/PRN, respectively [114]. Additionally, the proportion of patients gaining ≥15 letters from baseline was 34.5%, 30.2%, 36.1%, and 33.0% for the ranibizumab 0.5 mg/monthly; 0.5 mg/PRN; 2.0 mg/monthly, and 2.0 mg/PRN, respectively [114]. At month 24, the mean change in BCVA from baseline was +9.1 (0.5 mg monthly), +7.9 (0.5 mg PRN), +8.0 (2.0 mg monthly), and +7.6 (2.0 mg PRN) ETDRS letters, respectively [114]. The mean change in BCVA from month 12 to 24 was −1.0, −0.3, −1.2, and −1.0, for the ranibizumab 0.5 mg/monthly; 0.5 mg/PRN; 2.0 mg/monthly, and 2.0 mg/PRN, respectively. Moreover, the proportion of patients who gained ≥15 letters from baseline in BCVA at month 24 remained stable [115]. Thus, neither the higher 2 mg dose of ranibizumab nor the more strict monthly vs. PRN posology led to better visual outcomes than those achieved with monthly 0.5 mg ranibizumab [114,115].

The efficacy of a treat-and-extend (TREX) management strategy of intravitreal ranibizumab was assessed in treatment naïve NVAMD patients [116]. There were no significant differences in BCVA improvement between the monthly regimen (+9.2 letters) and the TREX strategy (+10.5 letters), *p* = 0.60. Nevertheless, the number of injections administered through the study follow-up was significantly lower in the TREX (10.1) than in the monthly (13.0) regime (*p* < 0.0001) [116].

In a separate study evaluating the efficacy and safety of ranibizumab administered in a TREX or in a monthly regimen, the TREX regimen was noninferior (*p* < 0.001) to the monthly regimen, with a mean BCVA improvement of 6.2 letters and 8.1 letters in the TREX and monthly groups, respectively [117].

The Canadian Treat-and-Extend Analysis Trial with ranibizumab (CANTREAT) was a 2-year, randomized, multicenter, and open-label study conducted on treatment-naive NVAMD patients [118,119]. At month 12, mean BCVA improvements of 8.4 ± 11.9 letters and 6.0 ± 11.9 letters were recorded in the TREX and monthly regimens, respectively [118]. Furthermore, at 24 months, mean BCVA improvement was not worse in the TREX treatment group (6.8 ± 14.1 letters) than in the monthly treatment group (6.0 ± 12.6 letters) [119].

Safety profile showed that incidence of adverse events, at both one- and two-year follow ups, were small (Table 5). With respect to ocular adverse events, eyes treated with ranibizumab more often developed cataracts compared with eyes in the control groups at both the one year (RR 1.48; 95% CI 0.83 to 2.66) and two year follow ups (RR 1.25; 95% CI 0.94 to 1.66) (See Table 5). Two eyes during the first year of ranibizumab injections and six more during the second year developed endophthalmitis, compared to no cases in the control eyes (Table 5). An overview of ocular adverse events is summarized in Table 4 and Table 5.

#### 3.1.4. Aflibercept

Aflibercept (EYLEA^®^; Bayer HealthCare, Berlin, Germany/Regeneron Pharmaceuticals Inc., Tarrytown, NY, USA) is a fusion protein (115 kDa) comprising the second Ig domain of human VEGFR1, the third Ig domain of human VEGFR2, and the Fc region of a human IgG1 [136,137,138].

VIEW 1 and VIEW 2 were parallel RCTs that included 2419 NVAMD patients in which monthly and every-2-month dosing of intravitreal aflibercept injection were compared with monthly ranibizumab injection. Subjects were randomly assigned to ranibizumab (0.5 mg) every 4 weeks (RQ4) or to 1 of 3 aflibercept doses: 0.5 mg every 4 weeks (0.5Q4), 2 mg every 4 weeks (2Q4), or 2 mg every 8 weeks after 3 monthly loading doses (2Q8) [120]. At week 52, the results of this study suggested that intravitreal aflibercept dosed monthly or every two months was not inferior to ranibizumab dosed monthly. The integrated analysis showed a mean BCVA improvement of +9.3, 8.7, 8.4, and 8.3 letters in the 2Q4, RQ4, 2Q8, and 0.5Q4 treatment groups, respectively [120]. At week 96 all the treatment regimens were equivalent. Mean BCVA gains ranged from 8.3 to 9.3 letters at week 52 and from 6.6 to 7.9 letters at week 96. The number of injections was mandated per protocol during the first 52 weeks of the study. From week 52 through 96, patients were treated PRN although injections were mandated at least every 12 weeks for both anti-VEGF agents (capped PRN posology). In a post hoc analysis, the number of injections from week 52 to week 96 was lower in the 2q4 and 2q8 groups vs. the Rq4 group: mean of aflibercept minus Rq4, −0.64 (95% CI, −0.89 to −0.40) for 2q4 and −0.55 (95% CI, −0.79 to −0.30) for 2q8 (*p* < 0.0001 for both) [139].

ALTAIR was a 96-week, randomized, open-label study that evaluated the efficacy and safety of intravitreal aflibercept (IVT-AFL) administered with two different TREX strategies, with either a 2-week (IVT-AFL-2W) or 4-week (IVT-AFL-4W) adjustment [121]. The mean (95% confidence interval) change in BCVA (ETDRS letters) from baseline to week 52 was 9.0 letters (6.4–11.5) and 8.4 letters (6.0–10.8) in the IVT-AFL-2W and IVT-AFL-4W groups, respectively [121]. From baseline to week 96, the mean change in BCVA was 7.6 (5.0–10.3) letters in the IVT-AFL-2W group and 6.1 letters (3.1–9.0) in the IVT-AFL-4W group [121].

The comparative effectiveness of intravitreal ranibizumab and aflibercept was assessed in a 12-month RCT [140]. Although at month 12 there was a difference in mean BCVA improvement in favor of ranibizumab (mean letter score difference, 2.3; 95% confidence interval: −0.1 to 4.7), this difference was not statistically significant (*p* = 0.06) [140].

Regarding safety, the most frequent adverse effect (at least 5% of patients treated with aflibercept) was conjunctival hemorrhage (26.7%), eye pain (10.3%), vitreous detachment (8.4%), cataract (7.9%), floating particles in the vitreous (7.5%), and an increase in intraocular pressure (7.2%). The serious adverse effects are rare and mostly related to injection procedure (Table 6).

#### 3.1.5. Brolucizumab

Brolucizumab (Beovu^®^, Novartis, Basel, Switzerland) is a low molecular weight, single-chain antibody fragment that targets all forms of VEGF-A with high affinity. It was developed for the treatment of NVAMD, and it is in pivotal studies for diabetic macular edema (DME) and macular edema secondary to retinal vein occlusion [141]. The efficacy and safety of brolucizumab was compared to that of aflibercept in a prospective, randomized, double-masked, and multicenter phase 2 study [142]. At week 40, there were no significant differences in mean BCVA change from baseline. As compared to aflibercept, a greater proportion of brolucizumab-treated eyes had resolved intraretinal and subretinal fluid [142].

The efficacy and safety of brolucizumab was compared to that of aflibercept in two similarly designed phase 3 non-inferiority (non-inferiority margin 4 ETDRS letters) studies (HAWK and HARRIER) [122]. Patients were randomized to intravitreal brolucizumab 3 mg (HAWK only) or 6 mg, or aflibercept 2 mg. Least-squares mean (standard error) BCVA change from baseline to week 48 was 6.1 (0.69), 6.6 (0.71, 6.8 (0.71), 6.9 (0.61), and 7.6 (0.61) in the brolucizumab 3 mg, brolucizumab 6 mg, and aflibercept 2 mg (HAWK study), and brolucizumab 6 mg and aflibercept 2 mg (HARRIER study), respectively [122]. Brolucizumab was not inferior to aflibercept in visual function at week 48, although anatomic outcomes favored brolucizumab over aflibercept [122].

Intraocular inflammation was identified in 50 (4.6%) of the brolucizumab-treated patients. Of those, 36 subjects (3.3%) had concomitant retinal vasculitis. Of the 36 subjects with intraocular inflammation and vasculitis, 23 subjects (2.1%) had concomitant vascular occlusion. The absolute risk of developing intraocular inflammation and losing 15 or more letters was 0.7% (8/1088). Eight (22%) of 36 patients with vasculitis lost ≥3 lines of vision, and five of 36 (14%) lost ≥6 lines vision over 2 years. Among the 23 patients with occlusive retinal vasculitis, 7 (30%) lost ≥3 lines of vision, and 5 (22%) lost ≥6 lines [122]. The American Society of Retina Specialists (ASRS) conducted a post-approval analysis of brolucizumab-associated intraocular inflammation cases with short-term follow-up of 26 eyes from 25 patients with retinal vasculitis (of which 85% were designated as occlusive). In this consecutive series, ≥3-line vision loss and ≥6-line vision loss was seen in 46% and 35% of eyes, respectively [143]. Despite the risk of vision loss associated with retinal vasculitis following brolucizumab injection, the overall rate of vision loss in the study population was not different when comparing the brolucizumab to aflibercept arms in HAWK and HARRIER.

The most common ocular adverse events were conjunctival hemorrhage (brolucizumab 3 and 6 mg; HAWK) and reduced visual acuity (aflibercept; HAWK, both treatments; HARRIER) [122]. Adverse events of interest included uveitis and iritis (2.2% for each) with brolucizumab 6 mg vs. 0.3% and 0%, respectively, with aflibercept in HAWK; corresponding rates in HARRIER were <1% in both arms [122]. The incidence of serious ocular adverse events was low in both trials; no event occurred in >1% of eyes [127]. Table 7 summarizes the main ocular adverse events reported by the HAWK and HARRIER in Study Eye.

## 4. Real-World Outcomes

Despite the fact that RCTs represent the best method for establishing the efficacy and safety for different interventions for a disease process, they are not free of weaknesses that limit their external validity [144]. The limitations of RCTs that underlie differences in outcomes observed within the trial vs. those in clinical practice include limited sample size, restricted enrollment criteria (e.g., inclusion of patients with few comorbidities), rigorously enforced follow-up and treatment schedules, assessment of limited outcomes, limited follow-up period, and ready access to the medication [145]. Real-world evidence allows us to assess whether results from RCTs can be applied to the general population. For example, the 1-year effect of intravitreal bevacizumab, ranibizumab, and aflibercept on VA was compared in a real-world setting in NVAMD patients included in the IRIS (Intelligent Research in Sight) Registry. The mean number of injections was higher in the ranibizumab (6.4 ± 2.4) and aflibercept groups (6.2 ± 2.4) compared to bevacizumab group (5.9 ± 2.4; *p* < 0.0001). Once adjusted by different covariates, no differences were observed in VA outcomes, suggesting clearly that all 3 drugs improve VA similarly over 1 year of monotherapy [146]. However, as with RCTs, real-life studies have inherent limitations and cofounders that should be taken into consideration when interpreting their results such as the lack of standardized assessment of visual acuity.

SEVEN-UP was a multicenter and noninterventional cohort study that evaluated the long-term outcomes of ranibizumab in patients enrolled in the ANCHOR, MARINA, and HORIZON trials [147]. According to the results of this study, approximately one third of patients undergoing ranibizumab therapy in the ANCHOR or MARINA study achieved good VA outcomes after 7 years. However, as compared to baseline, vision in one third declined by 15 letters or more [147]. It should be mentioned that many patients were withdrawn of the study, and only a few completed the follow-up period. Interestingly, NVAMD fellow eyes remained at risk for further vision decline in later years under management with low-frequency anti-VEGF therapy [148].

AURA was a retrospective, observational, and multicenter study that assessed management of patients with NVAMD receiving anti-VEGF treatment in clinical practice (149). This study found that the number of intravitreal anti-VEGF injections was lower in clinical practice than in RCTs, despite the observed initial improvement in VA. However, such improvement was not maintained over time [149]. Similarly, the Pan-American Collaborative Retina Study Group reported a retrospective case series that evaluated the long-term anatomical and functional outcomes of bevacizumab in patients with CNV secondary to AMD. This study found that the early VA gains achieved with intravitreal bevacizumab were not maintained after 5 years of follow-up [150]. Ozkaya et al. [151] found that approximately half of ranibizumab NVAMD treated patients remained stable after 5-years of follow-up.

The 5-year outcomes of the CATT study suggested that VA improvements achieved during the first 2 years were not maintained at 5 years. Nevertheless, it should be mentioned that at the 5-year visit, 50% of eyes had VA of 20/40 or better, confirming the positive long-term effect of anti-VEGF in NVAMD patients [28].

LUMINOUS was a prospective, multicenter, and observational 5-year study that assessed the effectiveness of ranibizumab 0.5 mg in either naïve or previously treated NVAMD patients [152]. A total of 6241 NVAMD naïve patients were included. The mean (standard deviation) VA improvement achieved at year-1 was 3.1 (16.51) letters. Data were stratified according to the baseline VA as <23 letters; 23 to <39 letters; 39 to <60 letters; 60 to <74 letters; and ≥74 letters. Based on this stratification, VA mean change was 12.6 (20.63), 6.7 (17.88), 3.6 (16.41), 0.3 (13.83), and −3.0 (11.82) letters at year 1, respectively. Although ranibizumab is an effective treatment in naïve NVAMD patients, baseline VA should be considered when interpreting visual outcomes [152].

According to the results of two meta-analyses, intravitreal ranibizumab for NVAMD prevents severe visual loss in real-world practice. However, patient outcomes are considerably worse than those reported in RCTs for both fixed and PRN regimens [27,153] (Figure 5). Among other factors, this difference in outcomes can be explained by the lower number of injections in real-world practice [152].

## 5. Limitations of Anti-VEGF Therapy

### 5.1. Non-Responders

Although anti-VEGF therapy has become the standard treatment for NVAMD [131], many patients do not respond adequately to this therapy or experience a slow loss of efficacy of anti-VEGF agents after repeated administration over time [154,155]. Resistance can occur at any time during the course of therapy [154,155]. Patients who do not respond adequately to anti-VEGF therapy can be classified into two groups, namely initial-non-responders (do not respond to the drug at the loading dose) and tachyphylactic patients (patients respond to the loading dose, but the response decreases gradually after repeated administrations) [155]. A framework of the potential causes of resistance to anti-VEGF therapy and their possible therapeutic approaches is summarized in Figure 6.

A retrospective study reported that 22 eyes (10.1%) were identified as initial non-responders to ranibizumab [156]. Another retrospective study found that 14 (22.2%) eyes (8 eyes receiving aflibercept and 6 ones receiving ranibizumab) were classified as non-responders, while 8 (12.7%) eyes were identified as tachyphylactic (7 eyes treated with aflibercept and one eye with ranibizumab) [157]. The results of a post-hoc subanalysis of the CATT study, which involved “pseudo-switching”, highlighted the importance of having a control group to evaluate the effect of switching therapy, an apparent improvement following a switch may be nothing more than a manifestation of regression to the mean (i.e., if a random variable is extreme on its first or first few measurements, it is likely to become closer to the mean or average on subsequent measurements) [158].

A post hoc subanalysis of the HARBOR trial, in which patients meeting specific futility criteria were maintained on the original therapy and followed, also demonstrated that apparent improvements in vision and optical coherence tomography (OCT) results attributed to switching may be nothing more than a manifestation of regression to the mean [159]. In fact, visual function and OCT improvements observed in these two “pseudo switching” trials were almost identical to that observed after actual switching studies [160]. Generally, apparent treatment futility can be a manifestation of inadequate treatment frequency as well as biological resistance based on the pharmacology of the applied treatment.

### 5.2. Cost

Monthly intravitreal injections are associated with a significant treatment burden for patients, caregivers, and physicians, often making such a regimen unachievable in clinical practice. According to the annual National Report of the Use of Drugs in Italy, a total of 137,387 patients with NVAMD received anti-VEGF therapy during the year 2018 [161]. Among them, 55.3% received treatment with ranibizumab; 25.3% were treated with aflibercept; 18.8% were treated with bevacizumab; and 0.5% received treatment with pegaptanib. The mean (standard deviation) number of injections during the first 12 months was 3.4 (3); 3.8 (84); 3.4 (3); and 2.6 (2), respectively [161]. As mentioned above, in real clinical practice patients usually received fewer anti-VEGF injections and achieved worse visual outcomes compared with patients receiving frequent therapy according to a fixed regimen in RCTs [27,153]. In fact, BCVA improvement correlated with treatment intensity at 1 year [162].

Additionally, differences in the overall treatment approach and how ophthalmologists use these anti-VEGF therapies have been identified. These differences may critically influence treatment effectiveness in real clinical practice [163]. According to the results of a meta-analysis, intravitreal aflibercept was associated with a higher overall treatment cost than ranibizumab (18,187 € vs. 17,168 €, respectively), although with an incremental gain in effectiveness (4.92 vs. 4.88 quality-adjusted life years, respectively) [163].

Despite the fact that bevacizumab is “off label” and not considered the standard of care for NVAMD in Europe (nor it is approved for treating NVAMD by regulatory agencies such as the US Food and Drug Administration [FDA] and the European Medicines Agency [EMA]), it has been identified as the most cost-effective treatment [164]. Table 8 shows a cost analysis comparison between bevacizumab, ranibizumab, and aflibercept.

## 6. NVAMD Management: Multi Target Approach

New approaches have been proposed for treating NVAMD. Indeed, VEGF is the main but not the only player in the NVAMD pathophysiology. However, over the past two years we have witnessed the failure of 3 phase II and III clinical trials that evaluated AMD treatments [165,166,167].

Pegpleranib (Fovista, Ophthotech, New York, NY, USA), a PDFG inhibitor, combined with bevacizumab, ranibizumab, or aflibercept did not result in benefit as measured by the mean change in BCVA at the 12-month time point [165]. The concomitant administration of squalamine (a supposed anti-VEGF booster and PDGF inhibitor) eye drops in combination with monthly intravitreal ranibizumab did not show significant advantages vs. intravitreal ranibizumab monotherapy [167]. Finally, nevascumab (Regeneron Inc., New York, NY, USA) is a fully human immunoglobulin G1 (IgG1) VelocImmune monoclonal antibody that selectively binds Ang2 with high affinity (K_d_ = 24 pM), blocks Ang2 binding to the Tie2 receptor, and does not bind to Ang1 [168]. The aim of nesvacumab is to rebalance the angiogenic Ang1-Ang-2 imbalance occurring in case of pathologic angiogenesis. Unfortunately, the results of a phase 2 study that evaluated the effect of adding nevascumab to intravitreal injections of aflibercept did not provide additional benefit, which might suggest a peripheral role of Ang-Tie pathway in comparison to VEGF-VEGFR2 one [169].

Nevertheless, Ang-Tie pathway blockade has recently come under additional scrutiny. Faricimab (Genentech/Roche, South San Francisco, CA) is the first bispecific antibody targeting both Ang2 and VEGF-A [170,171]. This single molecule has dual inhibition of both Ang2 and VEGF targets in NVAMD. Rather than monthly dosing, faricimab dosing ranges from 12 weeks to 16 weeks [170,171].

The AVENUE [172] and STAIRWAY [173] trials found faricimab to be noninferior to ranibizumab for treatment of NVAMD.

AVENUE was a head-to-head trial of 273 patients comparing 0.5 mg ranibizumab every 4 weeks to 1.6 mg or 6 mg faricimab given every 4 or 8 weeks in the treatment of NVAMD [172]. Over a follow-up period of 36 weeks, changes in BCVA, CMT, and CNV were comparable between treatment arms [172].

STAIRWAY is a 52-week study that assessed two extended dosing regimens of faricimab 6.0 mg given every 16 weeks or every 12 weeks, compared to ranibizumab 0.5 mg every four weeks [173]. At week 24 (three months after the last of four loading doses), patients randomized to faricimab every 16 weeks switched to 12-week dosing if they were shown to have active disease, per pre-defined criteria. At week 24, 65% (n = 36/55) of eyes treated with faricimab had no active disease, highlighting the potential of 16-week dosing in nearly two-thirds of patients. BCVA was fully maintained through to week 52 with both 16- and 12-week dosing regimens. Patients treated with faricimab dosed every 16 weeks had a mean improvement of 11.4 letters from baseline, compared to 10.1 letters in patients treated with faricimab injected every 12 weeks, and 9.6 letters in patients treated with monthly 0.5 mg ranibizumab. The three treatment regimens were similar in both the proportion of patients gaining more than 15 letters and avoiding a loss of more than 15 letters. Comparable reductions in central retina thickness were also observed [173]. The rates of ocular and systemic adverse events observed with faricimab were similar to the rates observed with ranibizumab. No new safety signal was observed [173].

Phase 3 studies with faricimab are ongoing.

OPT-302 (Opthea Ltd., Melbourne, Australia), a VEGF-C/D trap molecule, has been devised to be used in conjunction with existing standard-of-care anti-VEGF-A therapies [174]. The results of a phase 1/2a study that included 51 NVAMD patients showed that OPT-302 was noninferior as both monotherapy and in combination with ranibizumab [175].

Additionally, a phase 2b 24-week study, conducted on 366 NVAMD patients compared OPT-302 (0.5 mg and 2.0 mg) combined with ranibizumab vs. ranibizumab alone. The results showed that, as compared to 0.5 mg ranibizumab alone, combined therapy (OPT-302 + ranibizumab) provided a statistically significant improvement in visual function [174].

## 7. Designed Ankyrin Repeat Protein in NVAMD Treatment

Repeat proteins are the second most abundant protein classes functioning in protein-protein binding after immunoglobulins [176,177]. Proteins composed of ankyrin repeats (ARs), a motif of 33 amino acid residues, are important in modulating diverse cellular pathways necessary for the evolution of a more complicated multicellular organism [177]. Designed ankyrin repeat protein (DARPin) molecules, which are composed of natural ARs, are small, single-domain proteins that can selectively bind to a target protein with high affinity and specificity [178,179].

Besides their high selectivity and affinity, DARPin molecules also display remarkable stability that confers some advantages over currently available antibodies or antibody fragments as potential therapeutics. In addition, DARPin molecules can be specifically designed to modulate local or systemic pharmacokinetics [179,180]. DARPins have different clinical applications, including biochemical research, diagnosis, and therapy [181]. As a diagnostic tool, a specific DARPin for human epidermal growth factor receptor 2 was developed, demonstrating similar sensitivity but significantly higher specificity than a FDA-approved antibody [182]. Additionally, DARPins have been involved in preclinical tumor targeting, tumor killing, and as a vehicle in tumor therapy [183,184,185].

DARPins can be used in their monovalent form or conjugated to other chemical moieties, e.g., polyethylene glycol, for half-life modulation. DARPins can also be engineered to produce bi-specific or tri-specific compounds that bind different epitopes of the same target protein or two different target proteins [179]. A notable example of a tri-specific DARPin is MP0250, a drug candidate that can bind VEGF-A and hepatocyte growth factor (HGF) as well as one molecule of MP0250 binding two molecules of human serum albumin (HSA), to extend half-life [186]. MP0250 has successfully completed preclinical development and is currently in phase 2 clinical development for relapsed refractory multiple myeloma [187]. The strategy of double VEGF/HGF inhibition was suggested by the observation that resistance to VEGF inhibition can be induced by the activation of HGF/cMet signaling pathway [186]. In the same line of reasoning, bi-specific or tri-specific DARPins might be conceived and projected in the field of AMD, targeting the simultaneous inhibition of VEGF and PDGF or any other of the factors involved in the pathogenesis of disease, as described above. At present, we have MP0112, a DARPin engineered to bind VEGF-A with a *K*_d_ of 1–4 pmol/L [188,189]. Abicipar pegol (AGN-150998, MP0112, abicipar; Allergan plc/Molecular Partners) is a 14-kDa recombinant DARPin protein coupled to a 20-kDa polyethylene glycol (PEG) moiety to yield a 34-kDa molecule [190].

MP0112 was tested in a series of clinical trials for treatment of AMD [191] and DME [192]. The safety and effectivity of the DARPin MP0112 in treatment naïve NVAMD patients was primarily assessed in a phase 1/2, multicenter, open-label, and dose-escalation study [191]. Six cohorts of patients received a single MP0112 dose of 0.04 mg, 0.15 mg, 0.4 mg, 1.0 mg, 2.0 mg, or 3.6 mg, respectively. The BCVA remained stable and did not show any difference among the dosing cohorts during the study follow-up. CMT reduction was greater in those cohorts receiving 1.0 mg and 2.0 mg MP0112 (−95 µm and −111 µm, respectively), whereas patients who received 0.04 mg, 0.15 mg, and 0.4 mg of MP0112 had CMT reduction of −7 µm, −12 µm, and −62 µm, respectively. Treatment-related adverse events were reported in 13 (41%) patients, including one (3%) case of severe ocular inflammation with hypopyon in a patient who received 2.0 mg of MP0112 [191].

REACH was a phase 2, multicenter, randomized, and double-masked study designed to compare the efficacy and safety of abicipar vs. ranibizumab in patients with NVAMD [193]. The primary endpoint was the mean change in BCVA from baseline to week 16. Mean (standard deviation) change in BCVA at week 16 was +5.1(8.0), +7.6 (5.1), and 5.3 (11.1) letters in the abicipar 1 mg, abicipar 2 mg, and ranibizumab 0.5 mg, respectively, with no significant differences between them (Figure 7A). Additionally, mean (standard deviation) CMT reduction from baseline to week 16 was −154.0 (140.5) µm, −118.1 (97.1) µm, and −125.1 (92.6) µm in the abicipar 1 mg, abicipar 2 mg, and ranibizumab 0.5 mg cohorts, respectively, with no significant differences between them (Figure 7B).

Baseline mean (standard deviation) BCVA was 58.4 (13.5) letters, 58.5 (14.3) letters, and 60.4 (16.4) letters in the abicipar 1 mg, abicipar 2 mg, and ranibizumab 0.5 mg arms, respectively. Baseline mean (SD) CMT was 526.1 (165.1) µm, 466.0 (126.0) µm, and 463.3 (94.6) µm in the abicipar 1 mg, abicipar 2 mg, and ranibizumab 0.5 mg arms, respectively [193].

BAMBOO and CYPRESS were two phase 2, multicenter, randomized, and double-masked, 20-week studies conducted in Japan and the United States [194]. A total of 50 patients (25 in each study) received three monthly intravitreal injections of abicipar 1 mg or 2 mg or five monthly intravitreal injections of ranibizumab 0.5 mg (164). At week 16, mean (standard error of the mean) change in BCVA was +7.8 (2.7), +8.9 (2.9), and +17.4 (3.6) letters in the BAMBOO study and +4.4 (2.8) letters, +10.1 (3.3) letters, and +15.2 (3.0) letters in the CYPRESS study in the abicipar 1 mg, abicipar 2 mg, and ranibizumab arms, respectively. Mean (standard error of the mean) change in CMT from baseline to week 16, was −187.3 (46.1) μm, −196.5 (39.3) μm, and −230.4 (26.5) μm in the BAMBOO study and −106.5 (40.6) μm, −112.8 (53.7) μm, and −124.4 (22.1) μm in the CYPRESS study in the abicipar 1 mg, abicipar 2 mg, and ranibizumab arms, respectively [194].

CEDAR and SEQUOIA were two randomized, multicenter, double-masked, parallel-group, active-controlled, 2-year, and phase 3 clinical trials designed to compare the efficacy and safety of abicipar 2 mg (administered at baseline, week 4, week 8, and every 8 weeks thereafter through week 96) (Abicipar Q8); abicipar 2 mg (administered at baseline, week 4, week 12, and every 12 weeks thereafter through week 96) (Abicipar Q12); and ranibizumab 0.5 mg (administered at baseline and every 4 weeks through week 96) (ranibizumab Q4). The primary end point was stable vision (≤15 ETDRS letter loss) at week 52 [195]. The number of patients with stable vision at week 52 was 496 (93.2%), 481 (91.3%), and 564 (95.8%) in the abicipar Q8, abicipar Q12, and ranibizumab Q4 study groups, respectively. Mean CMT was reduced significantly in all the treatment groups from baseline to week 52 (see Table 9) [195]. At year 2 BCVA improvements achieved during the first year were maintained [196]. Moreover, only four intravitreal injections of abicipar were required to maintain the outcomes, as compared to monthly intravitreal ranibizumab injections [196].

Unfortunately results of phase 3 studies showed that abicipar-treated patients had higher risk of developing intraocular inflammation (IOI) and endophthalmitis compared with ranibizumab-treated patients. The incidence of drug-related ocular adverse events (AEs) was higher in the abicipar Q8 (16.8%) and abicipar Q12 (20.4%) groups than in the ranibizumab Q4 group (4.5%) because of the occurrence of IOI. Intraocular inflammation AEs in the study eye were reported for 96 patients (15.4%) in the abicipar Q8 group, 96 patients (15.3%) in the abicipar Q12 group, and 2 patients (0.3%) in the ranibizumab Q4 group. Uveitis, vitritis, and retinal vasculitis were the main IOI AEs. The occurrence of these complications led to both a higher rate of study discontinuations and a higher rate of vision loss in the abicipar groups. The most commonly reported treatment related adverse events have been summarized in Table 10.

As a result of the improvements in the manufacturing process, the incidence of intraocular inflammation (IOI) was 8.9 percent in the MAPLE study, which was lower than the rate observed in prior Phase 3 studies [197]. Most IOI events were assessed as mild to moderate in severity. The incidence of severe IOI was 1.6 percent with one reported case of iritis and one reported case of uveitis. There were no reported cases of endophthalmitis or retinal vasculitis in this study.

This IOI rate was lower than that observed in the abicipar arms in the SEQUOIA and CEDAR studies but higher than the rate observed in the ranibizumab arms in those studies [195].

In June 2020 the FDA has issued a Complete Response Letter to the Biologics License Application (BLA) for Abicipar pegol.

The letter from the FDA indicates that the rate of intraocular inflammation observed following administration of abicipar pegol 2mg/0.05 mL results in an unfavorable benefit-risk ratio in the treatment of NVAMD.

In July 2020 the company decided to withdraw application filings with both the EMA and the Japanese Regulatory Agency (PMDA) for abicipar and it is committed to working with the regulatory agencies to determine the appropriate next steps and discuss requirements for potential resubmissions.

Based on the latest news, the need for the new treatment options for NVAMD remains unmet.

## 8. Conclusions

AMD is a prevalent condition that constitutes one of the leading causes of irreversible visual impairment in industrialized countries. The overall prevalence is approximately 8.7%, although it varies significantly among different populations. From a clinical perspective, late-stage AMD can be classified atrophic and neovascular AMD. Although nonexudative AMD represents the most prevalent form, NVAMD is nonetheless a major cause of severe vision loss.

The exact pathogenesis of AMD has not been fully elucidated although the activation of a cascade of proinflammatory and proangiogenic responses, originating from damage to the choriocapillaris, the RPE and the outer retina may play an important role. The most studied factor related to neovascularization in AMD is VEGF-A. According to currently available scientific evidence, there are no clinically important differences in efficacy among the three anti-VEGF agents with which we have the greatest experience, i.e., aflibercept, bevacizumab, and ranibizumab. Moreover, the results achieved in clinical practice seem to be worse than those reported in clinical trials.

The next generation of VEGF inhibitors for ocular applications must offer significant advantages over the currently licensed products. Ideally, the administration should be simplified, and the frequency of intraocular injections should be reduced. DARPins directed against VEGF-A can specifically and potently inhibit angiogenesis and vascular leakage in vitro and in vivo. These treatments have been shown to be as effective as ranibizumab, but with greater durability, which may enhance patient compliance with needed injections and reduce the risk of serious adverse events such endophthalmitis, all of which may lead to better results in clinical practice (i.e., real world results) than have been achieved with aflibercept, bevacizumab, and ranibizumab. The differences in the structure of the paratope of DARPin broaden the spectrum of target molecules, while the ease of creating hybrid fusion proteins provides exceptional opportunities for creating bispecific and multivalent constructs. Faricimab, a bispecific antibody directed against VEFG-A and Ang2, also may offer greater durability than the currently used anti-VEGF agents.

A robust pipeline of drugs for NVAMD has merged over the past several years, including anti-VEGF A (abicipar pegol); anti-VEGF C & D (OPT-302); PDFG inhibitors (pegpleranib); Ang2 monoclonal antibodies (nevascumab); bispecific Ang2 and VEGF-A antibodies (faricimab); and complement inhibitors (avacincaptad pegol).

Although there have been new and promising developments in AMD treatment, unmet needs continue. They include the high number of intravitreal injections and indefinite evaluations; the duration of the treatment effects, progression of atrophy or subretinal fibrosis, and scar formation. Additionally, there is a need to identify alternative treatments that target other biological pathways, which might help to halt progression and restore vision in those affected by NVAMD.

## Figures and Tables

**Figure 1 ijms-21-08242-f001:**
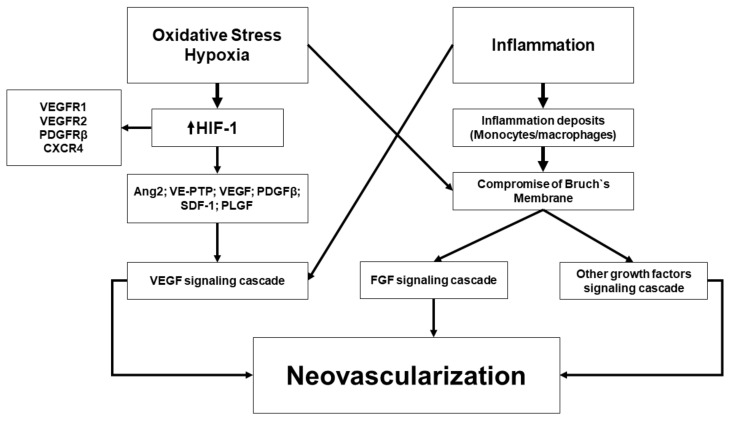
Overview of the pathophysiological leading to choroidal neovascularization. Adapted from Campochiaro [34] and Anderson et al. [41].

**Figure 2 ijms-21-08242-f002:**
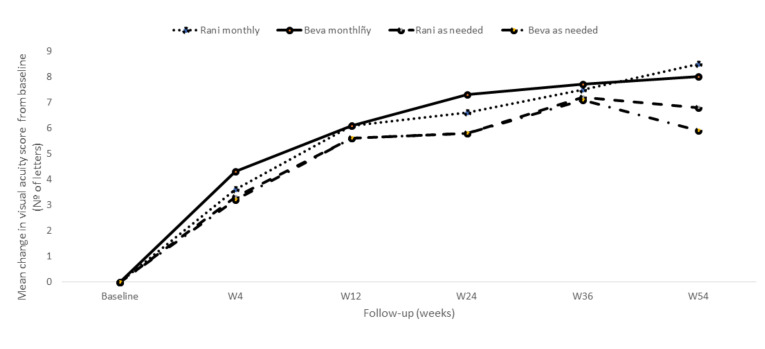
Mean change in the visual-acuity score during the first year of follow-up observed in the CATT study. Adapted from CATT Research Group et al. [100]. Rani: Ranibizumab; Beva: Bevacizumab.

**Figure 3 ijms-21-08242-f003:**
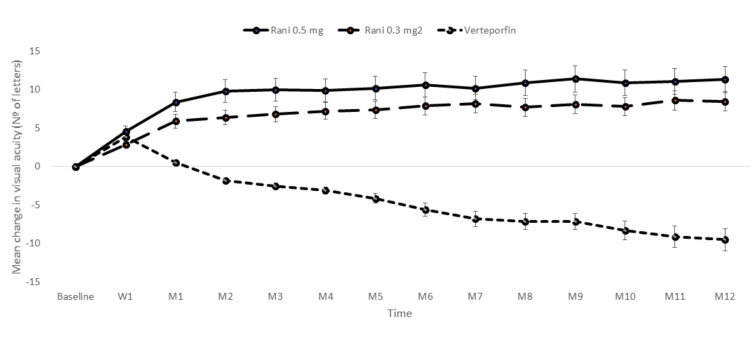
Mean (±SE) changes in the visual acuity from baseline through 12 months observed in the ANCHOR study. As compared to the verteporfin group, the mean visual acuity was significantly greater in each of the ranibizumab groups at each month during the first year (*p* < 0.001). Adapted from Brown et al. [90] and Brown et al. [93]. W: Week; M: Month. Rani: Ranibizumab; Beva: Bevacizumab.

**Figure 4 ijms-21-08242-f004:**
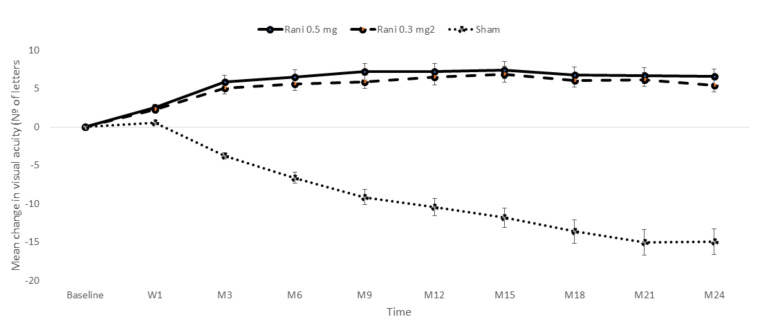
Mean (±SE) changes in the visual acuity from baseline through 24 months observed in the MARINA study. As compared to the sham injection group, the mean visual acuity was significantly greater in each of the ranibizumab groups at each month during the study follow-up (*p* < 0.001). At week 1, On day, *p* = 0.006 for patients receiving ranibizumab 0.3 mg and *p* = 0.003 for those receiving ranibizumab 0.5 mg. Adapted from Rosenfeld et al. [89]. W: Week; M. Month; Rani: Ranibizumab.

**Figure 5 ijms-21-08242-f005:**
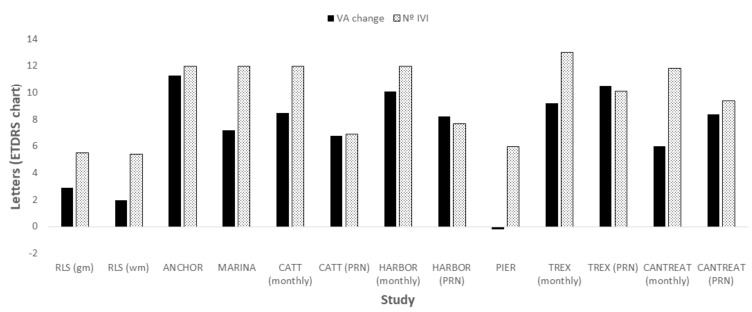
Mean change from baseline in visual acuity (VA) and number of intravitreal injections (IVI) in real-world studies and randomized controlled trials after 12 months of ranibizumab treatment in patients with neovascular age related macular degeneration. Real world studies results have been adapted from Chong [27] and Kim et al. [153]. RLS: Real-life studies; gm: grand means (mean of the means of the several studies); wm: weighted means (mean weighted against number of eyes per study).

**Figure 6 ijms-21-08242-f006:**
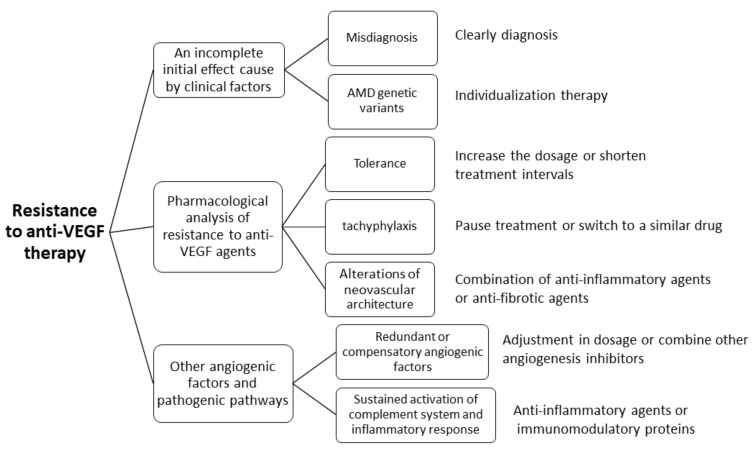
An overview of the different causes of resistance to anti-VEGF therapy and their possible therapeutic approaches. Adapted from Broadhead et al. [154] and Yang et al. [155]. Anti-VEGF: Vascular endothelial growth factor inhibitors; AMD: Age related macular degeneration.

**Figure 7 ijms-21-08242-f007:**
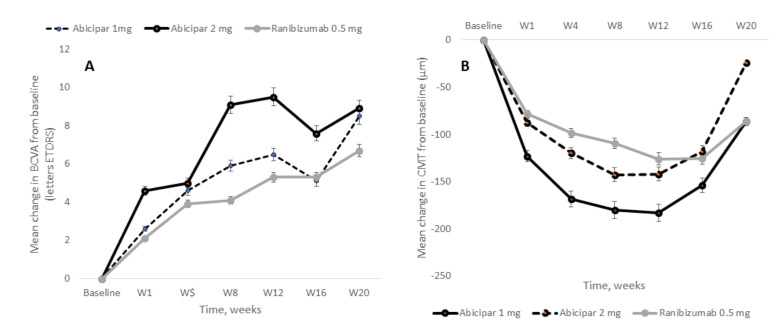
Mean changes in best corrected visual acuity (BCVA) (**A**) and central macular thickness (CMT) (**B**) from baseline in the modified intent-to-treat population of the REACH study. Adapted from Callanan et al. [193].

**Table 1 ijms-21-08242-t001:** Overview of the different studies evaluating the effects of vascular endothelial growth factor inhibitors (anti-VEGF) in patients with age related macular degeneration (AMD).

Study Treatment Period	Study Groups			
Pegaptanib vs. Control				
Gragoudas et al. [92] 2 years; re−randomized at end of first year	**Group I**	**Group II**	**Group III**	**Group IV**
	0.3 mg pegaptanib every 6 weeks	1.0 mg pegaptanib every 6 weeks	3.0 mg pegaptanib every 6 weeks	Sham every 6 weeks
Ranibizumab vs. Control				
ANCHOR [90,93] 2 years	**Group I**	**Group II**	**Group III**	
	0.3 mg ranibizumab monthly plus sham verteporfin PDT	0.5 mg ranibizumab monthly plus sham verteporfin PDT	Sham intravitreal injection plus verteporfin PDT	
MARINA [89,91] 2 years	**Group I**	**Group II**	**Group III**	
	0.3 mg ranibizumab monthly	0.5 mg ranibizumab monthly	Sham intravitreal injection monthly	-
PIER [94,95] 2 years	**Group I**	**Group II**	**Group III**	
	0.3 mg ranibizumab monthly for 3 months, then every 3 months	0.5 mg ranibizumab monthly for 3 months, then every 3 months	Sham intravitreal injection monthly for 3 months, then every 3 months	-
**Bevacizumab vs. Control**				
ABC [96,97] 1 year	**Group I**	**Group II**		
	1.25 mg bevacizumab given first 3 injections every 6 weeks, then as needed	Standard therapy (0.3 mg pegaptanib every 6 weeks, verteporfin PDT, or sham injection)	-	-
Sacu [98,99] 1 year	**Group I**	**Group II**		
	1.0 mg bevacizumab monthly for 3 months, then as needed	Verteporfin PDT plus same day 4 mg triamcinolone acetonide	-	-
**Bevacizumab vs. Ranibizumab**				
CATT [28,100,101] 2 years; re-randomized at end of first year	**Group I**	**Group II**	**Group III**	**Group IV**
	1.25 mg bevacizumab monthly for 1 year; at 1 year, re-randomization to ranibizumab monthly or variable dosing	0.5 mg ranibizumab monthly for 1 year; at 1 year, re-randomization to ranibizumab monthly or variable dosing	1.25 mg bevacizumab as needed after first injection for 2 years	0.5 mg ranibizumab as needed after first injection for 2 years
IVAN [102,103] 2 years; ongoing	**Group I**	**Group II**	**Group III**	**Group IV**
	1.25 mg bevacizumab monthly for 2 years	0.5 mg ranibizumab monthly for 2 years	1.25 mg bevacizumab monthly for 3 months, then as needed in 3-month cycles	0.5 mg ranibizumab monthly for 3 months, then as needed in 3-month cycles
Biswas et al. [104] 18 months	**Group I**	**Group II**		
	1.25 mg bevacizumab monthly for 3 months, then as needed	0.5 mg ranibizumab monthly for 3 months, then as needed	-	-
BRAMD [105] 1 year	Group I	Group II		
	1.25 mg bevacizumab monthly for 1 year	0.5 mg ranibizumab monthly for 1 year	-	-
GEFAL [106] 1 year	**Group I**	**Group II**		
	1.25 mg bevacizumab; maximum of 1 injection per month	0.5 mg ranibizumab; maximum of 1 injection per month	-	-
LUCAS [107,108] 1 year	**Group I**	**Group II**		
	1.25 mg bevacizumab; treat and extend protocol	0.5 mg ranibizumab; treat and extend protocol	-	-
	**Group I**	**Group II**		
MANTA [109] 1 year	1.25 mg bevacizumab monthly for 3 months, then as needed	0.5 mg ranibizumab monthly for 3 months, then as needed	-	-
SAVE-AMD [110] 1 year	**Group I**	**Group II**		
	1.25 mg bevacizumab at day 1 and at week 4, then as needed	0.5 mg ranibizumab at day 1 and at week 4, then as needed	-	-
Scholler et al. [111] 1 year	**Group I**	**Group II**		
	1.25 mg bevacizumab for 3 months, at 30-day intervals, then as needed	0.5 mg ranibizumab for 3 months, at 30-day intervals, then as needed	-	-
Subramanian et al. [112,113] 1 year	**Group I**	**Group II**		
	0.05 mL bevacizumab monthly for 3 months, then as needed	0.05 mL ranibizumab monthly for 3 months, then as needed	-	-

Abbreviations: ANCHOR: Anti-VEGF Antibody for the Treatment of Predominantly Classic Choroidal Neovascularization in Age-related Macular Degeneration; MARINA: Minimally Classic/Occult Trial of the Anti-VEGF Antibody Ranibizumab in the Treatment of Neovascular Age-Related Macular Degeneration; PIER: A Phase IIIb, Multicenter, Randomized, Double-Masked, Sham Injection-Controlled Study of the Efficacy and Safety of Ranibizumab in Subjects With Subfoveal Choroidal Neovascularization With or Without Classic CNV Secondary to Age-related Macular Degeneration; ABC: Th Avastin^®^ (Bevacizumab) in Choroidal Neovascularization Trial; CATT: Comparison of Age-related Macular Degeneration Treatment Trials; IVAN: A Randomized Controlled Trial of Alternative Treatments to Inhibit VEGF in Age-related Choroidal Neovascularisation; BRAMD: Comparison of Bevacizumab (Avastin^®^) and Ranibizumab (Lucentis^®^) in Exudative Age-related Macular Degeneration; GEFAL: French Evaluation Group Avastin^®^ vs. Lucentis^®^; LUCAS: Lucentis^®^ Compared to Avastin^®^ Study; MANTA: A Randomized Observer and Subject Masked Trial Comparing the Visual Outcome After Treatment With Ranibizumab or Bevacizumab in Patients With Neovascular Age-related Macular Degeneration Multicenter Anti-VEGF Trial in Austria; SAVE-AMD: Safety of VEGF Inhibitors in Age-Related Macular Degeneration; PDT: photodynamic therapy.

**Table 2 ijms-21-08242-t002:** Overview of the functional and anatomic results of the main studies evaluating the effect of vascular endothelial growth factor inhibitors (anti-VEGF) in patients with neovascular age related macular degeneration (NVAMD).

Study	Ref	Duration (w)	Regimen	N	BCVA (ETDRS Letters)		CMT (µm)	
					Baseline	Change	Baseline	Change
**ABC**	**[96]**	**54**	Beva 1.25 mg	65	50 (43–61) *	+7.0	328 (271–376) *	−93.5 (−144.5–−26)
			Control	66	53 (47–60) *	–9.4	330 (256–359) *	−55 (−150–7)
**CATT**	[100]	52	Beva 1.25 mg (monthly)	286	60.2 (13.1)	8.0 (1.0) **	463 (196)	−164 (181)
			Beva 1.25 (PRN)	300	60.4 (13.4)	5.9 (1.0) **	461 (175)	−152 (178)
			Rani 0.5 mg (monthly)	301	60.1 (14.3)	8.5 (0.8) **	458 (184)	−196 (176)
			Rani 0.5 mg (PRN)	298	61.5 (13.2)	6.8 (0.8) **	458 (193)	−168 (186)
**CATT**	[101]	104	Beva 1.25 mg (M/M)	135	60.2 (13.6)	7.8 (15.5)	462 (205)	−180 (196)
			Beva 1.25 (PRN/PRN)	270	60.6 (13.0)	5.0 (17.9)	459 (173)	−153 (189)
			Beva 1.25 mg (M/PRN)	131	60.4 (12.4)	NA	471 (185)	NA
			Rani 0.5 mg (M/M)	146	59.9 (14.2)	8.8 (15.9)	460 (190)	−190 (172)
			Rani 0.5 mg (PRN/PRN)	287	61.6 (13.1)	6.7 (14.6)	462 (195)	−166 (190)
			Rani 0.5 mg (MM/PRN)	138	60.9 (14.3)	NA	462 (184)	NA
**CATT**	[28]	5 years	Beva 1.25 mg	319	60.2 (24.1)	−2.1 (22.3)	460	278 ^†^
			Rani 0.5 mg	328	57.7 (42.1)	−4.5 (22.3)	466	289 ^†^
**ANCHOR**	[90]	52	Rani 0.3 mg	140	47.0 (13.1)	8.5 (14.6)	1.89 (1.44) ^‡^	0.36 (1.06)
			Rani 0.5 mg	140	47.1 (13.2)	11.3 (14.6)	1.79 (1.54) ^‡^	0.28 (1.29)
			Verteporfin	143	45.5 (13.1)	−9.5 (16.4)	1.88 (1.40) ^‡^	2.56 (3.09)
**ANCHOR**	[93]	104	Rani 0.3 mg	140	47.0 (13.1)	8.1 (16.2)	1.89 (1.44) ^‡^	0.52 (1.34)
			Rani 0.5 mg	140	47.1 (13.2)	10.7 (16.5)	1.79 (1.54) ^‡^	0.39 (1.34)
			Verteporfin	143	45.5 (13.1)	−9.8 (17.6)	1.88 (1.40) ^‡^	2.89 (3.33)
**MARINA**	[89]	52	Rani 0.3 mg	238	53.1 (12.9)	6.5	4.3 (2.5)	−0.27 (2.07)
			Rani 0.5 mg	240	53.7 812.8)	7.2	4.5 (2.6)	−0.01 (1.98)
			Sham	238	53.6 (14.1)	−10.4	4.4 (2.5)	1.91 (2.81)
**MARINA**	[91]	104	Rani 0.3 mg	238	53.1 (12.9)	5.4	4.3 (2.5)	−0.32 (2.41)
			Rani 0.5 mg	240	53.7 812.8)	6.6	4.5 (2.6)	−0.00 (2.04)
			Sham	238	53.6 (14.1)	−14.9	4.4 (2.5)	2.58 (2.81)
**HARBOR**	[114]	52	Rani 0.5 monthly	275	52.4 (13.3)	10.1	348.3 (146.3)	–172.0
			Rani 0.5 PRN	275	54.5 (11.7)	8.2	347.8 (143.8)	−161.2
			Rani 2.0 mg monthly	274	53.5 (13.1)	9.2	332.9 (138.7)	−163.3
			Rani 2.0 mg PRN	273	53.5 (13.2)	8.6	347.9 (142.9)	−172.4
**HARBOR**	[115]	104	Rani 0.5 monthly	275	52.4 (13.3)	9.1	348.3 (146.3)	−182.5
			Rani 0.5 PRN	275	54.5 (11.7)	7.9	347.8 (143.8)	−172.0
			Rani 2.0 mg monthly	274	53.5 (13.1)	8.0	332.9 (138.7)	−171.8
			Rani 2.0 mg PRN	273	53.5 (13.2)	7.6	347.9 (142.9)	−181.0
**TREX-AMD**	[116]	52	Rani 0.5 monthly	20	60.3 (2.4) **	9.2	533 (45) **	−173
			Rani 0.5 PRN	40	59.9 (2.4) **	10.5	489 (28) **	−246
**TREND**	[117]	104	Rani 0.5 monthly	327	60.6 (13.9)	7.9	497.7 (187.2)	−173.3
			Rani 0.5 PRN	323	59.5 (13.2)	6.6	504.0 (189.9)	−169.2
**CANTREAT**	[118]	52	Rani 0.5 monthly	293	59.5	6.0 (11.9)	374.2 (111.9)	NA
			Rani 0.5 PRN	287	58.9	8.4 (11.9)	382.5 (113.2)	NA
**CANTREAT**	[119]	104	Rani 0.5 monthly	293	59.5	6.0 (12.6)	374.2 (111.9)	NA
			Rani 0.5 PRN	287	58.9	6.8 (14.1)	382.5 (113.2)	NA
**VIEW 1**	[120]	52	Rani 0.5 mg Q4	304	54.0 (13.4)	8.1 (15.3)	315.3 (108.3)	−116.8 (109.0)
			Afli 0.5 mg Q4	301	55.6 (13.1)	6.9 (13.4)	313.2 (106.0)	−115.6 (104.1)
			Afli 2.0 mg Q4	304	55.2 (13.2)	10.9 (13.8)	313.6 (103.4)	−116.5 (98.4)
			Afli 2.0 mg Q8	301	55.7 (12.8)	7.9 (15.0)	324.4 (111.2)	−128.5 (108.5)
**VIEW 2**	[120]	52	Rani 0.5 mg Q4	291	53.8 (13.5)	9.4 (13.5)	325.9 (110.9)	−138.5 (122.2)
			Afli 0.5 mg Q4	296	51.6 (14.2)	9.7 (14.1)	326.5 (116.5)	−129.8 (114.8)
			Afli 2.0 mg Q4	309	52.8 (13.9)	7.6 (12.6)	334.6 (119.8)	−156.8 (122.8)
			Afli 2.0 mg Q8	306	51.6 (13.9)	8.9 (14.4)	342.6 (124.0)	−149.2 (119.7)
**ALTAIR**	[121]	52	IVT-AFL-2W	123	54.8 (13.1)	9.0	386.2 (159.2)	−134.4
			IVT-AFL-4W	123	55.3 (12.0)	8.4	370.3 (120.0)	−126.1
**ALTAIR**	[121]	96	IVT-AFL-2W	123	54.8 (13.1)	7.6	386.2 (159.2)	−130.5
			IVT-AFL-4W	123	55.3 (12.0)	6.1	370.3 (120.0)	−125.3
**HAWK**	[122]	48	Broluzizumab 3 mg	358	61.0 (13.6)	6.1 (0.7) **	466.6 (167.4)	−167.4 (6.9) **
			Broluzizumab 6 mg	360	60.8 (13.7)	6.6 (0.7) **	463.1 (166.2)	−172.8 (6.7) **
			Aflibercept 2 mg	360	60.0 (13.9)	6.8 (0.7) **	457.9 (146.4)	−143.7 (6.7) **
**HARRIER**	[122]	48	Broluzizumab 6 mg	370	61.5 (12.6)	6.9 (0.6) **	473.6 (171.4)	−193.8 (6.8) **
			Aflibercept 2 mg	369	60.8 (12.9)	7.6 (0.6) **	465.3 (151.2)	−143.9 (6.8) **

Note: * Median (interquartile range);** Standard error of the mean. ^†^ Mean macular thickness. ^‡^ Mean size of lesión. Abbreviations: Ref: Reference; w: Weeks; BCVA: Best-corrected visual acuity; EDTRS: Early Treatment of Diabetic Retinopathy Study; CMT: Central macular thickness; Rani: Ranibizumab; PRN: pro re nata; Q4: every 4 weeks; Q8: every 8 weeks; IVT-AFL-2W: Intravitreal aflibercept 2-week adjustment; IVT-AFL-4w: Intravitreal aflibercept 4 weeks adjustment.

**Table 3 ijms-21-08242-t003:** An overview of the ocular adverse events reported by age-related macular degeneration patients treated with pegaptanib. Adapted from Solomon et al. [124].

Ocular Adverse Event	0.3 Mg Pegaptanib N = 295	1.0 Mg Pegaptanib N = 301	3.0 Mg Pegaptanib N = 296	All Doses Pegaptanib N = 892	Control N = 298	RR (95% CI) All Doses vs. Control
Any eye disorder	9 (3.1%)	4 (1.3%)	10 (3.4%)	23 (2.6%)	2 (0.7%)	3.84 (0.91 to 16.20)
Endophthalmitis	6 (2.0%)	3 (1.0%)	3 (1.0%)	12 (1.3%)	0	8.37 (0.50 to 140.95)
Retinal detachment	1 (0.3%)	2 (0.7%)	2 (0.7%)	5 (0.6%)	0	3.68 (0.20 to 66.41)
Traumatic cataract	1 (0.3%)	2 (0.7%)	2 (0.7%)	5 (0.6 1%)	0	3.68 (0.20 to 66.41]
Retinal hemorrhage	1 (0.3%)	0	1 (0.3%)	2 (0.2%)	0	1.67 (0.08 to 34.77)
Vitreous hemorrhage	0	0	1 (0.3%)	1 (0.1%)	0	1.00 (0.04 to 24.59)
Uveitis	0	0	1 (0.3%)	1 (0.1%)	0	1.00 (0.04 to 24.59)
Elevated intraocular pressure	1 (0.3%)	0	0	1 (0.1%)	0	1.00 (0.04 to 24.59)
Papilledema	0	0	0	0	1 (0.3%)	0.11 (0.00 to 2.73)

RR: Relative risk; CI: Confidence interval.

**Table 4 ijms-21-08242-t004:** An overview of the ocular adverse events reported by age-related macular degeneration patients treated with bevacizumab. Adapted from Solomon et al. [124].

Serious Ocular Adverse Event ^a^	Studies Reporting Adverse Events ^1^	Bevacizumab		Ranibizumab		RR (95% CI) Bevacizumab vs. Ranibizumab
		Number with Event	Total Participants	Number with Event	Total Participants	
**Endophthalmitis ^a^**	CATT [100]; GEFAL [106]; LUCAS [107]	5 (0.5%)	1052	3 (0.3%)	1059	1.68 (0.40 to 7.00)
Retinal detachment	CATT [100]; GEFAL [106]	3 (0.4%)	832	0	838	7.05 (0.36 to 136.28)
Retinal pigment epithelial tear	CATT [100]; IVAN [102]; LUCAS [107]	4 (0.4%)	1102	3 (0.3%)	1134	1.37 (0.31 to 6.12)
Traumatic cataract	CATT [100]; GEFAL [106]; LUCAS [107]	1 (0.09%)	1128	2 (0.2%)	1152	0.51 (0.05 to 5.62)
Severe uveitis	CATT [100]; IVAN [102]	4 (0.5%)	882	1 (0.1%)	913	4.14 (0.46 to 36.97)
**Serious Ocular Adverse Event ^a^**	**Studies Reporting Adverse Events ^2^**	**Bevacizumab**		**Ranibizumab**		**RR (95% CI)** **Bevacizumab vs. Ranibizumab**
		**Number With Event**	**Total Participants**	**Number With Event**	**Total Participants**	
Endophthalmitis	CATT [101]	7	586	4	599	1.79 (0.53 to 6.08)
Traumatic cataract	IVAN [103]	1 (0.3%)	296	1 (0.3)	314	1.06 (0.07 to 16.88)
Severe uveitis	IVAN [103]	1 (0.3%)	296	0	314	3.18 (0.13 to 77.80)
Retinal detachment	IVAN [103]	0	296	1 (0.3)	314	0.35 (0.01 to 8.64)
Retinal pigment epithelial tear	IVAN [103]	1 (0.3%)	296	3 (1%)	314	0.35 (0.04 to 3.38)

^a^ Includes endophthalmitis and pseudo-endophthalmitis. ^1^. Adverse events at one year. ^2^. Adverse events at 2-years. RR: Relative risk; CI: Confidence interval.

**Table 5 ijms-21-08242-t005:** An overview of ocular adverse events reported by age-related macular degeneration patients treated with ranibizumab. Adapted from Solomon et al. [124].

Ocular Adverse Event ^a^	0.3 Mg Ranibizumab N = 196	0.5 Mg Ranibizumab N = 201	All Doses Ranibizumab N = 397	Control N = 206	RR [95% CI] All Doses vs. Control
Endophthalmitis	0	2 (1%)	2 (0.5%)	0	2.60 (0.13 to 53.92)
Retinal detachment	1 (0.5%)	0	1 (0.3%)	1 (0.5%)	0.52 (0.03 to 8.25)
Traumatic cataract	18 (9.2%)	22 (10.9%)	40 (10%)	14 (6.8%)	1.48 (0.83 to 2.66)
Retinal hemorrhage	2 (1%)	0	2 (0.5%)	2 (1%)	0.52 (0.07 to 3.66)
Vitreous hemorrhage	1 (0.5%)	0	1 (0.3%)	0	1.56 (0.06 to 38.13)
Uveitis	0	1 (0.5%)	1 (0.3%)	0	1.56 (0.06 to 38.13)
Elevated intraocular pressure (≥30 mmHg increase)	13 (6.6%)	17 (8.5%)	30 (7.6%)	7 (3.4%)	2.22 (0.99 to 4.98)
Ocular inflammation (trace to 4+)	21 (10.7%)	26 (12.9%)	47 (11.8%)	9 (4.4%)	2.71 (1.36 to 5.42)
**Ocular adverse event ^b^**	**0.3 mg ranibizumab** **n = 434**	**0.5 mg ranibizumab** **n = 440**	**All doses ranibizumab** **n = 874**	**Control** **n = 441**	**RR [95% CI]** **All doses vs. control**
Endophthalmitis	2 (0.5%)	6 (1.4%)	8 (0.9%)	0	8.59 (0.50 to 148.44)
Retinal detachment	2 (0.5%)	0	2 (0.2%)	2 (0.5%)	0.50 (0.07 to 3.57)
Traumatic cataract	65 (15%)	76 (17.3%)	141 (16.1%)	57 (12.9%)	1.25 (0.94 to 1.66)
Retinal hemorrhage	1 (0.2%)	0	1 (<0.1%)	1 (0.2%)	0.50 (0.03 to 8.05)
Vitreous hemorrhage	3 (0.7%)	1 (0.2%)	4 (<0.5%)	2 (0.5%)	1.01 (0.19 to 5.49)
Uveitis	3 (0.7%)	4 (0.9%)	7 (0.8%)	0	7.58 (0.43 to 132.36)
Elevated intraocular pressure (≥30 mmHg increase) ^c^	45 (15.2%)	61 (20.3%)	106 (17.8%)	11 (3,7%)	4.81 (2.63 to 8.81)
Ocular inflammation (1+ to 4+)	32 (7%)	30 (6.8%)	62 (7.1%)	8 (1.8%)	3.91 [1.89 to 8.09]

^a^ Adverse events up to 1 year. ^b^ Adverse events up to 2 years. ^c^ (n = 297 in 0.3 mg ranibizumab group, n = 300 in 0.5 mg ranibizumab group, and n = 298 in 0.3 mg control group). RR: Relative risk; CI: Confidence interval.

**Table 6 ijms-21-08242-t006:** An overview of the adverse events reported in the VIEW 1 and VIEW 2 studies. Adapted from Heier et al. [120] and Schmidt-Erfurth U et al. [139].

	VIEW 1		VIEW 2	
	Rani	AFL	Rani	AFL
	N = 304	N = 303	N = 291	N = 307
	n (%)	n (%)	n (%)	n (%)
AE				
Patients with ≥1 TEAEs, N (%)	290 (95.4)	289 (95.4)	250 (85.9)	277 (90.2)
Any ocular TEAE	263 (86.5)	257 (84.8)	210 (72.2)	220 (71.7
Study eye	246 (80.9)	238 (78.5)	187 (64.3)	198 (64.5)
Fellow eye	150 (49.3)	143 (47.2)	124 (42.6)	123 (40.1)
**SAE**				
Patients with ≥1 serious TEAEs, N (%)	68 (22.4)	56 (18.5)	35 (12.0)	48 (15.6)
Most common ocular SAEs				
Endophthalmitis	3 (1.0)	0	NR	NR
Reduced VA	2 (0.7)	0	1 (0.3)	5 (1.6)
Retinal hemorrhage	2 (0.7)	2 (0.7)	1 (0.3)	1 (0.3)
Most common injection-related ocular SAEs				
Endophthalmitis	3 (1.0)	0	NR	NR
**WDAE**				
WDAEs, N (%) (discontinuation from study)	4 (1.3)	4 (1.3)	2 (0.7)	9 (2.9)
Most common reasons				
Retinal hemorrhage	1 (0.3)	1 (0.3)	0 (0.0)	0 (0.0)
Endophthalmitis	1 (0.3)	0	NR	NR
**Deaths**				
Deaths, N (%)	5 (1.6)	8 (2.6)	2 (0.7)	2 (0.6)
**Notable Harms**				
Retinal detachment	NR	NR	15 (5.2)	12 (3.9)
ATE	5 (1.6)	12 (4.0)	0	0

AFL: Aflibercept; ATE: Arterial thrombotic event; NR: Not reported; Rani: Ranibizumab; SAE: Serious adverse event; TEAE: Treatment-emergent adverse event; VA: Visual acuity; WDAE: Withdrawal due to adverse event.

**Table 7 ijms-21-08242-t007:** Overview of the ocular adverse events (AE) reported by the HAWK and HARRIER in Study Eye. Adapted from Dugel et al. [122].

Adverse Event, n (%)	HAWK			HARRIER	
	Brolucizumab 3 mg (N = 358)	Brolucizumab 6 mg (N = 360)	Aflibercept 2 mg (N = 360)	Brolucizumab 6 mg (N = 370)	Aflibercept 2 mg (N = 369)
Patients with ≥1 event	175 (48.9)	179 (49.7)	170 (47.2)	122 (33.0)	119 (32.2)
Conjunctival hemorrhage	30 (8.4)	23 (6.4)	20 (5.6)	7 (1.9)	12 (3.3)
Visual acuity reduced	23 (6.4)	19 (5.3)	24 (6.7)	20 (5.4)	20 (5.4)
Vitreous floaters	24 (6.7)	18 (5.0)	11 (3.1)	11 (3.0)	3 (0.8)
Eye pain	21 (5.9)	16 (4.4)	15 (4.2)	10 (2.7)	12 (3.3)
Dry eye	11 (3.1)	14 (3.9)	15 (4.2)	8 (2.2)	6 (1.6)
Retinal hemorrhage	10 (2.8)	13 (3.6)	16 (4.4)	5 (1.4)	2 (0.5)
Retinal pigment epithelial tear	5 (1.4)	12 (3.3)	4 (1.1)	6 (1.6)	4 (1.1)
Vitreous detachment	16 (4.5)	10 (2.8)	13 (3.6)	7 (1.9)	5 (1.4)
Eye irritation	8 (2.2)	10 (2.8)	8 (2.2)	3 (0.8)	1 (0.3)
Intraocular pressure increased	11 (3.1)	9 (2.5)	8 (2.2)	12 (3.2)	9 (2.4)
Posterior capsule opacification	5 (1.4)	9 (2.5)	7 (1.9)	5 (1.4)	1 (0.3)
Uveitis	5 (1.4)	8 (2.2)	1 (0.3)	3 (0.8)	0
Blepharitis	4 (1.1)	8 (2.2)	7 (1.9)	8 (2.2)	3 (0.8)
Iritis	1 (0.3)	8 (2.2)	0	0	1 (0.3)
Cataract	10 (2.8)	7 (1.9)	8 (2.2)	4 (1.1)	12 (3.3)
Visual field defect	7 (2.0)	7 (1.9)	3 (0.8)	1 (0.3)	0
Conjunctivitis	2 (0.6)	7 (1.9)	3 (0.8)	10 (2.7)	3 (0.8)
Vision blurred	11 (3.1)	6 (1.7)	5 (1.4)	1 (0.3)	2 (0.5)
Visual impairment	10 (2.8)	6 (1.7)	10 (2.8)	0	2 (0.5)
Punctate keratitis	5 (1.4)	6 (1.7)	8 (2.2)	1 (0.3)	3 (0.8)
Corneal abrasion	5 (1.4)	5 (1.4)	8 (2.2)	0	1 (0.3)
Lenticular opacities	6 (1.7)	0	3 (0.8)	8 (2.2)	7 (1.9)

**Table 8 ijms-21-08242-t008:** Differences in effectiveness and costs between vascular endothelial growth factor inhibitors (anti-VEGF), per patient during the first year of treatment. Adapted from van Asten et al. [164].

	Bevacizumab *	Ranibizumab	Aflibercept
Cost, €			
Mean 95% CI	27,087 (22,818 to 31,789)	33,137 (28,883 to 37,926)	31,119 (26,979 to 35,766)
Differences in cost, €			
Mean	N.A.	6,050	4,032
Mean effectiveness, QALY			
Mean 95% CI	0.69 (0.66 to 0.73)	0.69 (0.66 to 0.73)	0.71 (0.67 to 0.74)
ICER, Δ€/ΔQALY (€)	N.A.	Dominated ^†^	278,099

* Bevacizumab as comparator.^†^ It is costlier, but does not yield health benefit. CI: Confidence Interval; NA: Not applicable; QALY: quality-adjusted life years; ICER: incremental costs-effectiveness ratio.

**Table 9 ijms-21-08242-t009:** Overview of the functional and anatomic results of abicipar.

Study	Ref	Duration (w)	Regimen	N	BCVA (ETDRS Letters)		CRT (µm)	
					Baseline	Change	Baseline	Change
**Souied et al.**	[191]	16	Abicipar 0.04 mg	9	N.A.	N.A.	352 (107.8) *	7
			Abicipar 0.15 mg	7				−12
			Abicipar 0.4 mg	6				−62
			Abicipar 1.0 mg	6				−95
			Abicipar 2.0 mg	4				−111
			Abicipar 3.6 mg	0				N.A.
**REACH**	[193]	20	Abicipar 1 mg	25	58 (13)	8.5 (8.1)	526 (165)	−86.2 (124.4)
			Abicipar 2 mg	23	59 (14)	8.9 (5.5)	466 (126)	−24.3 (54.1)
			Ranibizumab 0.5 mg	16	60 (16)	6.7 (7.7)	463 (95)	−86.1 (113.4)
**BAMBOO**	[194]	20	Abicipar 1 mg	10	54.3 (24−72) ^†^	7.8 (2.7) ^‡^	475.1 (296−840) ^†^	−187.3 (46.1)
			Abicipar 2 mg	10	58.5 (27−75) ^†^	8.9 (2.9) ^‡^	438.7 (288−591) ^†^	−196.5 (39.3)
			Ranibizumab 0.5 mg	5	55.8 (47−70) ^†^	17.4 (3.6) ^‡^	470.0 (363−538) ^†^	−230.4 (26.5)
**CYPRESS**	[194]	20	Abicipar 1 mg	10	55.2 (33−70)^†^	4.4 (2.8) ^‡^	443.8 (285−643) ^†^	−106.5 (40.6)
			Abicipar 2 mg	10	59.0 (42−74) ^†^	10.1 (3.3) ^‡^	383.8 (278−792) ^†^	−112.8 (53.7)
			Ranibizumab 0.5 mg	5	57.6 (30−72) ^†^	15.2 (3.0) ^‡^	348.8 (313−395) ^†^	−124.4 (22.1)
**CEDAR**	[195]	52	Abicipar Q8	265	56.8 (12.8) **	22.6% ^⁋^	382.5 (130.7) **	−134.6 (4.8)
			Abicipar Q12	262	56.5 (12.7) **	19.2% ^⁋^	378.3 (121.4) **	−141.0 (4.8)
			Ranibizumab Q4	290	56.8 812.4) **	27.2% ^⁋^	380.3 (125.5) **	−143.2 (4.7)
**SEQUOIA**	[195]	52	Abicipar Q8	267	56.8 (12.8) **	28.2% ^⁋^	382.5 (130.7) **	−141.8 (4.2)
			Abicipar Q12	265	56.5 (12.7) **	24.2% ^⁋^	378.3 (121.4) **	−139.4 (4.2)
			Ranibizumab Q4	299	56.8 812.4) **	26.7% ^⁋^	380.3 (125.5) **	−145.3 (4.1)
**Khurana RN ^1^**	[196]	104	Abicipar Q8	630	56.8 (12.8) **	7.8	382.5 (130.7) **	N.A.
			Abicipar Q12	628	56.5 (12.7) **	6.1	378.3 (121.4) **	
			Ranibizumab Q4	630	56.8 812.4) **	8.5	380.3 (125.5) **	

Note: * Mean of the overall study population. The study does not provide such information per groups. ^†^ Range ^‡^ Mean (standard error of the mean) at week 16. ** Intent to treat population of the two studies were pooled. Proportion of patients who gained ≥15-letter in BCVA from baseline to week 52. ^1^ 2-year results of the CEDAR and SEQUOIA studies. The data of the two studies were pooled for analysis. Abbreviations: Ref: Reference; w: Weeks; BCVA: Best-corrected visual acuity; EDTRS: Early Treatment of Diabetic Retinopathy Study; CMT: Central macular thickness; N.A.: Not applicable; Q8: Abicipar administered at baseline week 4–week 8, and every 8 weeks thereafter through week 96; Q12: Abicipar administered at baseline week 4—week 12, and every 12 weeks thereafter through week 96; Q4: Ranibizumab administered at baseline and every 4 weeks through week 96.

**Table 10 ijms-21-08242-t010:** Ocular adverse events reported in different studies evaluating designed ankyrin repeat proteins (DARPins).

Study	Ref	TRAE	Treatment Regime		
**REACH**	[193]		***Abicipar 1 mg (*n *= 25)***	***Abicipar 2 mg (*n *= 23)***	***Ranibizumab 0.5 mg (*n *= 16)***
		Overall incidence ^a^	11 (44.0)	7 (30.4)	5 (31.3)
		Vitreous floaters	3 (12.0)	1 (4.3)	1 (6.3)
		IOI *	3 (12.0)	2 (8.7)	0 (0.0)
		Vitreous detachment	2 (8.0)	2 (8.7)	0
		Retinal hemorrhage	3 (12.0)	0	2 (12.5)
		Eye pain	1 (4.0)	2 (8.7)	1 (6.3)
		Conjunctival hemorrhage	2 (8.0)	0	0
		Macular scar	0	0	2 (12.5)
**BAMBOO and CYPRESS**	[194]	**TRAE**	***Abicipar 1 mg (*n *= 20)***	***Abicipar 2 mg (*n *= 20)***	***Ranibizumab 0.5 mg (*n *= 10)***
		Conjunctival hemorrhage	4 (20.0)	0 (0.0)	1 (10)
		Dry eye	2 (10.0)	1 (5.0)	0 (0.0)
		Cataract	1 (5.0)	0 (0.0)	1 (10.0)
		Eye pain	0 (0.0)	1 (5.0)	1 (10.0)
		FBS	0 (0.0)	1 (5.0)	1 (10.0)
		Increased IOP	0 (0.0)	0 (0.0)	2 (20.0)
		Iritis	1 (5.0)	1 (5.0)	0 (0.0)
		Edema peripheral	0 (0.0)	1 (5.09	1 (10.0)
		Pneumonia	0 (0.0)	1 (5.0)	1 (10.0)
		Vitreous floaters	1 (5.0)	1 (5.0)	0 (0.0)
		Vitreous opacities	1 (5.0)	1 (5.0)	0 (0.0)
		Vitritis	1 (5.0)	1 (5.0)	0 (0.0)
**CEDAR and SEQUOIA**	[195]	**TRAE**	**Abicipar Q8 (n = 625)**	**Abicipar Q12 (n = 626)**	**Ranibizumab Q4 (n = 625)**
		Any	203 (32.5)	233 (37.2)	152 (24.3)
		Study drug related	105 (16.8)	128 (20.4)	28 (4.5)
		Study procedure related	142 (22.7)	168 (26.8)	143 (22.9)
		Serious	43 (6.9)	42 (6.7)	2 (0.3)
		IOI	99 (15.8)	96 (15.6)	1 (0.2) **

^a^ Patients with 1 or more AEs in the study eye. * It included iritis, uveitis, iridocyclitis, and vitritis. ** Reported after patient received an injection of abicipar from an incorrect study medication kit. Ref: Reference; TRAE: Treatment related adverse event; IOI: Intraocular inflammation; FBS: Foreign body sensation; IOP: Intraocular pressure.

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
