# Peer review of "Neovascular Age-Related Macular Degeneration: Therapeutic Management and New-Upcoming Approaches"

_ijms, 2020, doi:10.3390/ijms21218242_

Round 1

Reviewer 1 Report

Table 2. Some headings and text was not properly oriented – perhaps a landscape orientation would be better?

Figs 2, 3 & Fig 4. Legend is not clear as hard to identify the prn groups in Fig 2 eg. Rani as needed, Beva as needed, please make the legend easier to identify.

Line 676 states “Central macula thickness (CMT) reduction was …” without giving detail. Please specify micron of reduction.

Author Response

Response to the reviewers, manuscript ID IJMS-972515.

I have considered the comments made by the reviewers and I hope that the paper improves greatly thanks to these comments.

As this paper is not intended exclusively for ophthalmologists and due to it would be published within a monographic edition, we considered it appropriate to draw up an introductory overview as exhaustive as possible. Obviously, we made any effort to improve the weaknesses highlighted by the referees.

I am sending you the revised manuscript and the rebuttal letter providing a point-by-point response to each of the numbered reviewer comments.

#Reviewer: 1

Comments to the Author

Table 2. Some headings and text were not properly oriented – perhaps a landscape orientation would be better?

The section 2 “Classification of AMD” was removed following indications of reviewer 2.

Figs 2, 3 & Fig 4. Legend is not clear as hard to identify the prn groups in Fig 2 eg. Rani as needed, Beva as needed, please make the legend easier to identify.

Figure 2: The following information was added to the legend:

Rani: Ranibizumab; Beva: Bevacizumab.

Figure 3: The following information was added to the legend:

Rani: Ranibizumab; Beva: Bevacizumab.

Figure 4: The following information was added to the legend:

Rani: Ranibizumab

Line 676 states “Central macula thickness (CMT) reduction was …” without giving detail. Please specify micron of reduction.

In the former lines 676-678, new ones 716-718 was written: “Central macular thickness (CMT) reduction was greater in those cohorts receiving 1.0 mg and 2.0 mg MP0112 (-95 µm and -111 µm, respectively), whereas patients who received 0.04 mg, 0.15 mg, and 0.4 mg of MP0112 had CMT reduction of -7 µm, -12 µm, and -62 µm, respectively”.

The reduction was specified in the first version of the manuscript.

English language and style are fine/minor spell check required

Language was carefully reviewed and edited.

Reviewer 2 Report

Ricci et al. propose a timely and update review about the therapy of AMD with "designed ankyrin repeat proteins" (DARPins). The review process is very well conducted and summarized, and the topic is very relevant for the therapeutic perspective that DARPins have on one of the commonest eye diseases.

My major concerns regard the background sections of the manuscript: from chapter 1 to 4. I find the following weakness: are logically poorly organized (it is not clear the aim of the sections and the relationship among them), tables are not commented in the text but often just reported without comment on its meaning for the general topic (for example that on risk factors), and in general it is not clear the relevance of such a broad discussion for a relative specific sub-topic. I had the impression that summarizing such a large knowledge in so little space, the text became too cryptic for the unexpert reader and too general for the expert. I suggest restructuring these sections to make them more focused on the main topic of the review.  

Author Response

Response to the reviewers, manuscript ID IJMS-972515.

I have considered the comments made by the reviewers and I hope that the paper improves greatly thanks to these comments.

As this paper is not intended exclusively for ophthalmologists and due to it would be published within a monographic edition, we considered it appropriate to draw up an introductory overview as exhaustive as possible. Obviously, we made any effort to improve the weaknesses highlighted by the referees.

I am sending you the revised manuscript and the rebuttal letter providing a point-by-point response to each of the numbered reviewer comments.

#Reviewer 2

Ricci et al. propose a timely and update review about the therapy of AMD with "designed ankyrin repeat proteins" (DARPins). The review process is very well conducted and summarized, and the topic is very relevant for the therapeutic perspective that DARPins have on one of the commonest eye diseases.

Thank you very much; we highly appreciated the reviewer comment.

My major concerns regard the background sections of the manuscript: from chapter 1 to 4. I find the following weakness: are logically poorly organized (it is not clear the aim of the sections and the relationship among them), tables are not commented in the text but often just reported without comment on its meaning for the general topic (for example that on risk factors), and in general it is not clear the relevance of such a broad discussion for a relative specific sub-topic. I had the impression that summarizing such a large knowledge in so little space, the text became too cryptic for the unexpert reader and too general for the expert. I suggest restructuring these sections to make them more focused on the main topic of the review.  

Different changes were done in the following sections:

Introduction:

  1. table 1 was removed.

Prevalence information was moved to the beginning of the section.

The following sentence was removed: Geographic atrophy also can cause significant loss of vision [1,2,4], and its prevalence increases substantially among persons over the age of 80 years [9,10].

The following sentences were added in the introduction section:

“Wet AMD has been subclassified as Type 1 “occult”/polypoidal choroidal vasculopathy, Type 2 “classic”, and Type 3 “retinal angiomatous proliferation”. The angiogenesis and increased vascular permeability seen in wet AMD is driven partly by upregulation of vascular endothelial growth factor (VEGF) [24]”.

“Angiogenesis is a multistep, tightly regulated process that is controlled by a dynamic balance of positive and negative factors. It has been proposed that the main type responsible for angiogenesis is VEGF-A, interacting with VEGF receptor (VEGFR) 1 and 2 [25]”.

“Despite having drugs that achieve positive results in clinical trials, real-world results differ significantly [27]. Moreover, clinical trials have also shown poor long-term results of anti-VEGF [28]”.

“Newer treatment strategies are, therefore, needed to close the gap between clinical trial results and real-world settings. Currently, there are several therapeutic options that are being investigated – designed ankyrin repeat proteins (DARPINs), brolucizumab, use of port-delivery systems (PDS), and bispecific antibodies”.

Section 2: “Classification of AMD”

This section was removed.

Section 3: pathophysiology

This section was maintained without changes (only language editing), because we think that may be helpful for the reader.

Section 4: Treatment Strategies of NVAMD

4.1. Vascular Endothelial Growth Factor Inhibitors

The following paragraph was added: “Many different studies have evaluated the efficacy and safety of anti-VEGF in NVAMD patients. The results of these studies point on the same general direction, indicating a significant reduction in central macular thickness (CMT) and a visual acuity improvement”.

4.1.1. Pegaptanib

The following sentence was added: Regarding safety, incidence of ocular adverse events in the pegaptanib groups was greater than in the control group (see table 3). Moreover, subjects in the pegaptanib groups were more likely to have a serious systemic adverse event than participants in the control one.

4.1.2. Bevacizumab

The following information was added: “Regarding safety, at one year, in the CATT 2011 [100], IVAN [102], and GEFAL 2013 [106], less than 1% of participants had endophthalmitis, retinal detachment, retinal pigment epithelial tear, traumatic cataract, or uveitis. At two years, less than 1% of participants had endophthalmitis, retinal detachment, retinal pigment epithelial tear, traumatic cataract, or uveitis [101,103]”.

4.1.3 Ranibizumab

The following information was added: “Safety profile showed that incidence of adverse events, at both one- and two-year follow ups, were small (table 5). With respect to ocular adverse events, eyes treated with ranibizumab more often developed cataracts compared with eyes in the control groups at both the one year (RR 1.48; 95% CI 0.83 to 2.66) and two year follow ups (RR 1.25; 95% CI 0.94 to 1.66) (See table 5). Two eyes during the first year of ranibizumab injections and six more during the second year developed endophthalmitis, compared to no cases in the control eyes (table 5)”.

4.1.5. Brolucizumab

The following information was added: “The most common ocular adverse events were conjunctival hemorrhage (brolucizumab 3 and 6 mg; HAWK) and reduced visual acuity (aflibercept; HAWK, both treatments; HARRIER) [122]. Adverse events of interest included uveitis and iritis (2.2% for each) with brolucizumab 6 mg versus 0.3% and 0%, respectively, with aflibercept in HAWK; corresponding rates in HARRIER were <1% in both arms [122]. The incidence of serious ocular adverse events was low in both trials; no event occurred in >1% of eyes [122]”.

Round 2

Reviewer 2 Report

The article has greatly improved after the revision.